# Discrete Inversion: A Controllable Latent Space for Masked Generative Models

## Abstract

Discrete diffusion models have achieved notable success in tasks like image generation and masked language modeling, yet they face limitations in controlled content editing. This paper introduces **Discrete Inversion**, the first approach to enable precise inversion for discrete diffusion models, including multinomial diffusion and masked generative models. By recording noise sequences and masking patterns during the forward diffusion process, Discrete Inversion facilitates accurate reconstruction and controlled edits without the need for predefined masks or attention map manipulation. We demonstrate the effectiveness of our method across both image and text domains, evaluating it on models like VQ-Diffusion, Paella, and RoBERTa. Our results show that Discrete Inversion not only preserves high fidelity in the original data but also enables flexible and user-friendly editing in discrete spaces, significantly advancing the capabilities of discrete generative models.

## 1 Introduction

Diffusion models have emerged as a powerful class of generative models, demonstrating remarkable success in image synthesis (Ho et al., 2020; Song et al., 2020; Nichol & Dhariwal, 2021). These models learn to generate data by iteratively denoising samples from a simple noise distribution, effectively reversing a diffusion process that gradually corrupts data. Broadly, diffusion models can be categorized into continuous and discrete types.

Continuous diffusion models operate in continuous spaces, leveraging stochastic differential equations (SDEs) or their deterministic counterparts, ordinary differential equations (ODEs), to model the forward and reverse diffusion processes (Song et al., 2020; 2021). Advances such as flow matching (Lipman et al., 2022; Liu et al., 2022; Albergo & Vanden-Eijnden, 2022; Albergo et al.) have enhanced their efficiency and flexibility. These models have been successfully applied in various domains, including image editing (Meng et al., 2021; Avrahami et al., 2022; Mokady et al., 2022; Han et al., 2024; Zhang et al., 2023b), medical imaging (He et al., 2023), and solving inverse problems (Chung et al., 2022; Stathopoulos et al., 2024). In image editing, continuous diffusion models enable controlled manipulation of images while preserving consistency with the underlying data distribution. A key capability enabling this is *inversion*—the process of reversing the diffusion model to recover the original noise vector or latent representation that could have generated a given data sample. Two main inversion approaches exist: deterministic inversion using ODEs (e.g., DDIM Inversion (Song et al., 2021)) and stochastic inversion by recording noise sequences (e.g., CycleDiffusion (Wu & De la Torre, 2022), DDPM Inversion (Dhariwal & Nichol, 2021)).

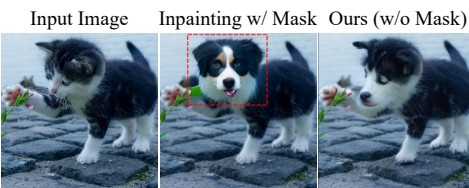

Input Image  Inpainting w/ Mask  Ours (w/o Mask)

Black and white ~~cat~~ dog on floor

Figure 1: Illustration of the limitation of masked inpainting method. Here, we want to change the cat to a dog. Inpainting with masked generation inadvertently modifies the orientation of the head, resulting in a less favourable result. With our discrete inversion, we are able to edit the image while preserving other properties of the object being edited. This is achieved by injecting the information from the input image into the logit space. Dotted red box indicates the mask, base model is Paella (Rampas et al., 2022).

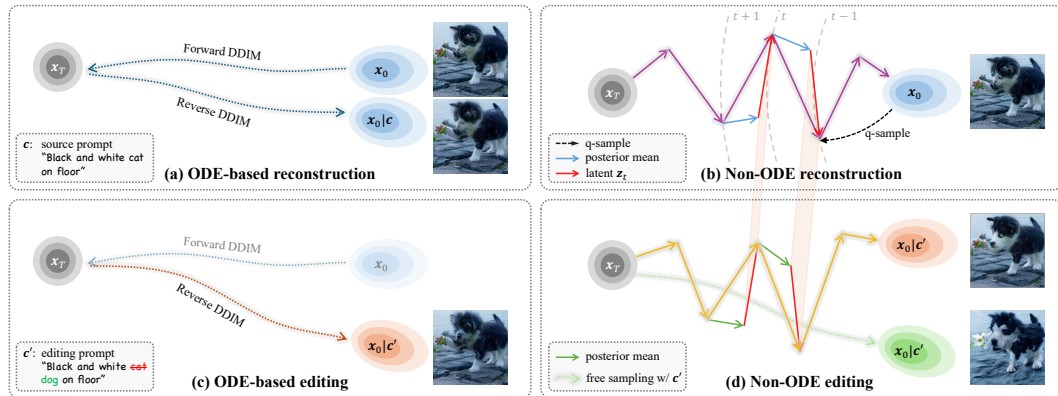

Figure 2: Here we demonstrate the two types of reconstruction and editing paradigms, namely ODE-based and Non-ODE based. (a,c) shows the ODE-based editing and reconstructions, while it provides accurate editing and reconstruction performances, it highly depends on the underlying ODE trajectory, which is not feasible in the discrete diffusion. However, the Non-ODE editing samples a trajectory by directly adding noise to $x_0$ and record the difference between the predicted $x_{t-1}$ and the sampled $x_{t-1}$ as indicated in the red arrow. In this way, we are able to reconstruct/edit the image without the strong condition of having an underlying ODE.

Discrete diffusion models are designed for inherently discrete data such as text or image tokens (Esser et al., 2021b). They adapt the diffusion framework to discrete spaces by defining appropriate transition kernels that corrupt and restore discrete data (Hoogeboom et al., 2021; Austin et al., 2021; Gu et al., 2022). Prominent examples include multinomial diffusion (Hoogeboom et al., 2021; Gu et al., 2022), D3PM (Austin et al., 2021), and masked generative models like MaskGIT (Chang et al., 2022), Muse (Chang et al., 2023). Despite their success in generation tasks, discrete diffusion models face limitations in controlled content editing. For instance, masked generative models achieve image editing through masked inpainting, where regions are masked and regenerated based on new conditions. However, this approach lacks the ability to inject information from the masked area into the inpainting process, limiting fine-grained control over the editing outcome, as illustrated in Figure 1.

Moreover, existing ODE-based inversion techniques developed for continuous diffusion models are not directly applicable to discrete diffusion models due to inherent differences in data representation and diffusion processes. This gap hinders the ability to perform precise inversion and controlled editing in discrete spaces. To address this challenge, we propose **Discrete Inversion** (Discrete Inversion for Controllable Editing), the first inversion algorithm for discrete diffusion models to the best of our knowledge. Our method extends the stochastic inversion approach to discrete diffusion models, including both multinomial diffusion and masked generative models. The core idea is to record the noise sequence needed to recover a stochastic trajectory in the reverse diffusion process. Specifically, given an artificial trajectory where latent states have low correlation, we fit reverse sampling steps to this trajectory and save the residuals between targets and predictions. This process *imprints* the information of the original input data into the recorded residuals. During editing or inference, the residuals are added back, allowing us to inject and control the amount of information introduced into the inference process.

Our approach enables accurate reconstruction of the original input data and facilitates controlled editing without the need for predefined masks or attention map manipulation. It provides a flexible framework for fine-grained content manipulation in discrete spaces, overcoming the limitations of existing methods. We validate the effectiveness of Discrete Inversion through extensive experiments on both image and text modalities. We evaluate our method on models such as VQ-Diffusion (Gu et al., 2022), Paella (Rampas et al., 2022), and RoBERTa (Liu et al., 2019), demonstrating its versatility across different types of discrete generative models. Additionally, we introduce a novel text-editing dataset to further showcase our method's capabilities and to facilitate future research in this area. Contributions of this paper can be summarized as follows:

- We introduce Discrete Inversion, an inversion algorithm for discrete diffusion models, including multinomial diffusion and masked generative models. By recording and injecting noise sequences or masking patterns, Discrete Inversion enables accurate reconstruction and controlled editing of discrete data without predefined masks or attention manipulation.

- We validate the effectiveness of Discrete Inversion through comprehensive experiments on both image and text modalities, demonstrating its versatility across different types of discrete generative models.

- We show that our approach can transform a model primarily trained for understanding tasks, such as RoBERTa, into a competitive generative model for text generation and editing, illustrating the potential for extending discrete diffusion models to new applications.

## 2 RELATED WORK

**Discrete Diffusion.** D3PM (Austin et al., 2021) and Multinomial Diffusion (Hoogeboom et al., 2021) spearheaded the study of diffusion processes in discrete spaces by developing a corruption mechanism for categorical data. Following those works, Esser et al. (2021a) and Gu et al. (2022) introduced the VQ-GAN as a way to discretize the image into tokens. Additionally, Campbell et al. (2022) proposed discrete diffusion models with continuous time, while Lou et al. (2023) extended score matching (Song & Ermon, 2019) to discrete spaces by learning probability ratios. Gat et al. (2024) proposed discrete flow matching to extend the flow matching to discrete space.

Masked Sequence Modeling has been widely used in representation learning for natural language processing. In models like BERT (Devlin et al., 2018) and RoBERTa (Liu et al., 2019), masked tokens (`[MASK]`) are predicted based on the surrounding context, excelling in text completion and embedding representation learning. Wang & Cho (2019) first interpreted the BERT model as a Markov Random Field and studied its generative perspective. Mask-Predict (Ghazvininejad et al., 2019) proposed a similar iterative remask-and-repredict algorithm for machine translation. For image generation, Paella (Rampas et al., 2022) adapts this approach for text-conditional image generation by renoising tokens instead of masking (like in MaskGIT (Chang et al., 2022) and Muse (Chang et al., 2023)). These models can be viewed as a special case of discrete diffusion models by introducing an *absorbing state* (Austin et al., 2021). The inference process of these models is typically heuristic and follows a renoise-and-repredict scheme.

**Diffusion inversion.** Diffusion inversion aims to find an encoding or latent representation of the input signal that can be used to reconstruct the original data. Traditional approaches to diffusion inversion are based on neural ODEs (Chen et al., 2018), such as DDIM inversion (Song et al., 2021) and flow matching (Lipman et al., 2022; Liu et al., 2022), where deterministic trajectories are used for inversion. Another class of methods focuses on stochastic differential equations (SDEs) (Song et al., 2020), including models like CycleDiffusion (Wu & De la Torre, 2022) and DDPM Inversion (Huberman-Spiegelglas et al., 2024), which rely on tracking noise or residuals along a stochastic path to recover the input. Our approach generalizes the concept of DDPM Inversion by extending it to discrete diffusion models, enabling effective inversion in both continuous and discrete settings.

**Inversion-based image editing.** DDIM inversion (Song et al., 2021) has served as a foundational technique for various diffusion-based image editing approaches. In many image editing tasks, DDIM-type methods are often employed alongside guidance techniques like Prompt-to-Prompt (Hertz et al., 2022), which manipulate cross-attention maps, as well as self-attention maps, as demonstrated by approaches like Plug-and-Play (Tumanyan et al., 2023), TF-ICON (Lu et al., 2023), and StyleAligned (Hertz et al., 2024). On the other hand, DDPM inversion-based approaches (Huberman-Spiegelglas et al., 2024) are known for their user-friendly nature, as they typically do not require complex attention map manipulations. These approaches are also versatile and can integrate with semantic guidance techniques, such as SEGA Brack et al. (2023) and LEDITS++ Brack et al. (2024), enabling broader applicability. To address issues such as inaccurate reconstruction and error accumulation, Null-text Inversion (Mokady et al., 2022) introduces test-time optimization of null embeddings, ensuring the reconstruction trajectory aligns more closely with the DDIM inversion path. Negative-prompt Inversion (Miyake et al., 2023; Han et al., 2024) further improves time efficiency by providing a closed-form solution to an approximate inversion problem, reducing computational costs while maintaining competitive reconstruction quality.

## 3 METHODS

### 3.1 PRELIMINARIES

Denoting $\boldsymbol{x}_0 \in \{1, \ldots, K\}^D$ as a data point of dimension $D$. We use $\boldsymbol{v}(x_t^{(i)})$ to denote the one hot column vector representation of the $i$-th entry of $\boldsymbol{x}_t$. To simplify notation, in the following we drop index $i$ and any function that operates on vector $\boldsymbol{x}_t$ is populated along its dimension. Diffusion model defines a Markov chain $q(\boldsymbol{x}_{1:T}|\boldsymbol{x}_0) = \Pi_{t=1}^T q(\boldsymbol{x}_t|\boldsymbol{x}_{t-1})$ that gradually add noise to the data $\boldsymbol{x}_0$ for $T$ times so that $\boldsymbol{x}_T$ contains little to no information. Discrete diffusion model (Hoogeboom et al., 2021; Austin et al., 2021; Gu et al., 2022) proposed an alternative likelihood-based model for categorical data, and defines the forward process following:

$$q(x_t|x_{t-1}) = \text{Cat}(\boldsymbol{v}(x_t); \boldsymbol{p} = \boldsymbol{Q}_t \boldsymbol{v}(x_{t-1})). \tag{1}$$

where $\boldsymbol{Q}_t$ is the transition matrix between adjacent states following mask-and-replace strategy, and $\text{Cat}(\cdot; \boldsymbol{p})$ denotes the categorical distribution with probabilities $\boldsymbol{p}$. The posterior distribution given $x_0$ has a closed-form solution,

$$q\left(x_{t-1}|x_t, x_0\right) = \frac{(\boldsymbol{Q}_t^\top \boldsymbol{v}(x_t)) \odot (\overline{\boldsymbol{Q}}_{t-1} \boldsymbol{v}(x_0))}{\boldsymbol{v}(x_t)^\top \overline{\boldsymbol{Q}}_t \boldsymbol{v}(x_0)}. \tag{2}$$

where $\overline{\boldsymbol{Q}}_t = \boldsymbol{Q}_t \cdots \boldsymbol{Q}_1$ is the cumulative transition matrix. The details of $\boldsymbol{Q}_t$ and $\overline{\boldsymbol{Q}}_t$ are given in the supplementary materials. The inference process is as below:

$$\boldsymbol{\pi}_\theta(x_t, t) = p_\theta\left(x_{t-1}|x_t\right) = \sum_{\tilde{x}_0=1}^K q\left(x_{t-1}|x_t, \tilde{x}_0\right) p_\theta\left(\tilde{x}_0|x_t\right), \tag{3}$$

with $p_\theta(\tilde{x}_0|x_t)$ is parameterized by a neural network. We gradually denoise from $x_T$ to $x_0$ using 3. For numerical stability, the implementation uses log space instead of probability space. Masked generative models can be viewed as a special case of multinomial diffusion models with an additional *absorbing* state (or the [MASK] state). Its training objective can be viewed as a reweighted ELBO (Bond-Taylor et al., 2022).

### 3.2 DISCRETE INVERSION

**Non ODE-based inversion.** ODE-based generative models, such as DDIM and flow matching, define an ODE trajectory. Due to the deterministic nature of ODEs, inversion can be achieved by solving the ODE using the Euler method in forward direction, ensuring reconstruction based on the inherent properties of the ODE. In contrast, another line of research focuses on SDE-based models, such as CycleDiffusion (Wu & De la Torre, 2022) and DDPM Inversion (Huberman-Spiegelglas et al., 2024). Broadly speaking, these approaches ensure reconstruction by recording the noises or residuals that are required to reproduce the stochastic trajectory. CycleDiffusion records the Gaussian noise $\boldsymbol{z}_t$ during sampling from posterior $p(\boldsymbol{x}_{t-1}|\boldsymbol{x}_t, \boldsymbol{x}_0 = \boldsymbol{x}_0)$ and injects information of the input signal by feeding the true $\boldsymbol{x}_0$. DDPM Inversion, on the other hand, incorporates information into $\boldsymbol{z}_t$ by fitting the reverse process into an artificial stochastic trajectory obtained by independent q-sample. For both CycleDiffusion and DDPM Inversion, the key idea is to utilize the Gaussian reparameterization trick, $x = \mu + \sigma z \Leftrightarrow x \sim \mathcal{N}(x; \mu, \sigma^2)$, and keeping track of the "noise" that could have generated the sample from mean. For discrete diffusion models, we utilize the Gumbel-Max trick (Maddison et al., 2014; Jang et al., 2016), $x = \arg\max \log(\boldsymbol{\pi}) + \boldsymbol{g} \Leftrightarrow x \sim \text{Cat}(x; \boldsymbol{\pi})$. Figure 2 provides an intuition of the proposed method.

**Inverting masked generative models.** For masked generative modeling, the stochastic trajectory $\{\boldsymbol{x}_t\}$ is constructed according to the specific inference algorithm of the model in use. For example, in Paella Rampas et al. (2022), the masking is *inclusive*, meaning that as the time step $t$ increases, the set of masked tokens grows. In contrast, the Unleashing Transformer Bond-Taylor et al. (2022) employs *random* masking at each step, where masks are generated independently using the q-sample function. Without loss of generality, we define a denoiser function $\mathcal{D}_\theta$ (parameterized by $\theta$). This denoiser outputs the *logits* of the predicted unmasked data given the noisy tokens $\boldsymbol{x}_t$. Since the inference of DDPM or multinomial diffusion is different from masked modeling, where $x_{t-1}$ is *not* sampled from a posterior given $x_t$. Instead, $x_t$ is obtained from sampled $\hat{x}_{0|t}$ by re-noising. Since

the categorical sampling happens at sampling from the denoiser's prediction, we therefore define an corresponding latent sequence:

$$\hat{\boldsymbol{y}}_{0|t} = \log(p_\theta(\boldsymbol{x}_0|\boldsymbol{x}_t)) = \mathcal{D}_\theta(\boldsymbol{x}_t, t) \tag{4}$$

$$\boldsymbol{z}_t := \boldsymbol{y}_0 - \hat{\boldsymbol{y}}_{0|t}. \tag{5}$$

With our proposed latent space, accurate reconstruction is guaranteed. However, for editing tasks, this level of precision may not be ideal if the latent variable $\boldsymbol{z}_t$ dominates the generation process. The detailed algorithm is given in Algorithm 1.

To provide more flexibility, we introduce the hyperparameters $\tau$, $\lambda_1$, and $\lambda_2$, which allow for finer control over the editing process. Specifically, $\tau$ represents the starting (and largest) timestep at which the editing process begins, while $\lambda_1$ controls the amount of information injected from the original input, and $\lambda_2$ governs the introduction of random noise.

---

**Algorithm 1** Discrete Inversion for Masked Generative Modeling

**Inversion:**
1: $\boldsymbol{y}_0 \leftarrow \mathcal{D}(\boldsymbol{x}_0, \boldsymbol{c}, t = 0)$
2: Sample noise token map $\boldsymbol{n}$
3: **for** $t$ from 1 to $T$ **do**
4: $\quad \boldsymbol{m}_t \leftarrow \text{GenerateMask}(t)$  $\triangleright$ Sampling masks according to inference algorithm
5: $\quad \boldsymbol{x}_t \leftarrow \boldsymbol{x}_0 \odot (\mathbf{1} - \boldsymbol{m}_t) + \boldsymbol{n} \odot \boldsymbol{m}_t$
6: $\quad \hat{\boldsymbol{y}}_{0|t} \leftarrow \mathcal{D}_\theta(\mathbf{x}_t, \boldsymbol{c}, t = t)$
7: $\quad \boldsymbol{z}_t \leftarrow \boldsymbol{y}_0 - \hat{\boldsymbol{y}}_{0|t}$
8: **end for**
**Sampling:**
9: **for** $t$ from $\tau$ to 1 **do**
10: $\quad \hat{\boldsymbol{y}}_{0|t} \leftarrow \mathcal{D}_\theta(\mathbf{x}_t, \boldsymbol{c}', t = t)$
11: $\quad \boldsymbol{g} \sim \text{Gumbel}(\mathbf{0}, \boldsymbol{I})$
12: $\quad \tilde{\boldsymbol{y}}_0 \leftarrow \hat{\boldsymbol{y}}_{0|t} + \lambda_1 \cdot \boldsymbol{z}_t + \lambda_2 \cdot \boldsymbol{g}$
13: $\quad \tilde{\boldsymbol{x}}_0 \leftarrow \arg\max \tilde{\boldsymbol{y}}_0$
14: $\quad \boldsymbol{x}_{t-1} \leftarrow \tilde{\boldsymbol{x}}_0 \odot (\mathbf{1} - \boldsymbol{m}_{t-1}) + \boldsymbol{n} \odot \boldsymbol{m}_{t-1}$
$\quad \quad \triangleright$ Re-noise
15: **end for**
16: Return $\boldsymbol{x}_0$.

**Algorithm 2** Discrete Inversion for Multinomial Diffusion

**Inversion:**
1: **for** $t$ from 1 to $T$ **do**
2: $\quad \boldsymbol{x}_t \sim q(\boldsymbol{x}_t|\boldsymbol{x}_0)$ $\triangleright$ Independent q-sample using 6
3: $\quad \boldsymbol{y}_t \leftarrow \log(\text{onehot}(\boldsymbol{x}_t))$
4: **end for**
5: **for** $t$ from $T$ to 1 **do**
6: $\quad \hat{\boldsymbol{y}}_{t-1} \leftarrow \log(\boldsymbol{\pi}_\theta(\boldsymbol{x}_t, \boldsymbol{c}, t))$  $\triangleright$ Log posterior using 3
7: $\quad \boldsymbol{z}_t \leftarrow \boldsymbol{y}_{t-1} - \hat{\boldsymbol{y}}_{t-1}$
8: **end for**
**Sampling:**
9: **for** $t$ from $\tau$ to 1 **do**
10: $\quad \hat{\boldsymbol{x}}_0 \leftarrow p_\theta(\boldsymbol{x}_0|\mathbf{x}_t = \arg\max \boldsymbol{y}_t)$
11: $\quad \boldsymbol{g} \sim \text{Gumbel}(\mathbf{0}, \boldsymbol{I})$
12: $\quad \boldsymbol{y}_{t-1} \leftarrow \log(q(\boldsymbol{x}_{t-1}|\boldsymbol{x}_t, \hat{\boldsymbol{x}}_0; \boldsymbol{c}')) + \lambda_1 \cdot \boldsymbol{z}_t + \lambda_2 \cdot \boldsymbol{g}$  $\triangleright$ Using Gumbel trick
13: **end for**
14: Return $\boldsymbol{x}_0 = \arg\max \boldsymbol{y}_0$.

---

**Noise injection.** We discuss three strategies as follows:

*Linear.* This is a natural form inspired by the Gumbel-Max trick: thinking of $\lambda_1 \cdot \boldsymbol{z}$ as a correction term, then $\log(\boldsymbol{\pi}) + \lambda_1 \cdot \boldsymbol{z}$ is the corrected logit and $\lambda_2$ is the inverse of temperature of the logit to control the sharpness of the resulting categorical distribution, as

$$\arg\max\left(\log(\boldsymbol{\pi}) + \lambda_1 \cdot \boldsymbol{z} + \lambda_2 \cdot \boldsymbol{g}\right)$$

$$= \arg\max\left(\frac{1}{\lambda_2}\left(\log(\boldsymbol{\pi}) + \lambda_1 \cdot \boldsymbol{z}\right) + \boldsymbol{g}\right), \quad \lambda_2 > 0.$$

$\lambda_1$ then controls how much correction we would like to introduce in the original logit.

*Variance preserving.* From another perspective, $\boldsymbol{z}$ is the artificial "Gumbel" noise that could have been sampled to realize the target tokens. Then, if we treat $\boldsymbol{z}$ as Gumbel noise and want to perturb it with random Gumbel noise, addition does not result in a Gumbel distribution. One way is to approximate this sum with another Gumbel distribution. If $G_1 \sim \text{Gumbel}(\mu_1, \beta_1)$, $G_2 \sim \text{Gumbel}(\mu_2, \beta_2)$ and $G = \lambda_1 G_1 + \lambda_2 G_2$, then the moment matching *Gumbel approximation* for $G$ is

$$\text{Gumbel}(\mu_G, \beta_G), \quad \text{with}$$

$$\beta_G = \sqrt{\lambda_1^2 \beta_1^2 + \lambda_2^2 \beta_2^2},$$

$$\mu_G = \lambda_1 \mu_1 + \lambda_2 \mu_2 + \gamma(\lambda_1 \beta_1 + \lambda_2 \beta_2 - \beta_G),$$

where $\gamma \approx 0.5772$ is the Euler-Mascheroni constant. We consider the *variance preserving* form:

$$\tilde{\boldsymbol{y}} = \log(\boldsymbol{\pi}) + \sqrt{\lambda_1} \cdot \boldsymbol{z} + \sqrt{\lambda_2} \cdot \boldsymbol{g}, \;\; \lambda_1 + \lambda_2 = 1.$$

*Max.* The third way is inspired by the property of Gumbel distribution (Wikipedia contributors, 2024), that if $G_1, G_2$ are iid random variables following Gumbel$(\mu, \beta)$ then $\max\{G_1, G_2\} - \beta \log 2$ follows the same distribution. We also consider the *max* function for noise injection:

$$\tilde{\boldsymbol{y}} = \log(\boldsymbol{\pi}) + \max\{\lambda_1 \cdot \boldsymbol{z}, \lambda_2 \cdot \boldsymbol{g}\}.$$

We empirically find that *linear* strategy gives best results.

**Inverting multinomial diffusion** is more straightforward given its inference is similar to DDPM. We start by sampling a stochastic trajectory, $\{\boldsymbol{x}_t\}$, a sequence of independent q-sample's from $q(\boldsymbol{x}_t|\boldsymbol{x}_0)$ (we populate the following sampling operation along the dimension of $\boldsymbol{x}_t$),

$$x_t = \arg\max\left(\log(q(x_t|x_0)) + \boldsymbol{g}\right), \;\; \text{with} \tag{6}$$

$$q(x_t|x_0) = \text{Cat}(x_t; \boldsymbol{p} = \overline{\boldsymbol{Q}}_t \boldsymbol{v}(x_0)) \;\; \text{and} \;\; \boldsymbol{g} \sim \text{Gumbel}(\boldsymbol{0}, \boldsymbol{I}).$$

Note that here we use the Gumbel softmax trick (Jang et al., 2016), which is equivalent to sampling from categorical distribution $q(x_t|x_0)$.

$$\boldsymbol{y}_{t-1} = \log(\text{onehot}(\boldsymbol{x}_{t-1})), \;\; \text{and} \tag{7}$$

$$\hat{\boldsymbol{y}}_{t-1} = \log(\boldsymbol{\pi}_\theta(\boldsymbol{x}_t, t)), \tag{8}$$

$$\boldsymbol{z}_t := \boldsymbol{y}_{t-1} - \hat{\boldsymbol{y}}_{t-1} \tag{9}$$

Note that here the latent $\boldsymbol{z}_t \in \mathbb{R}^{D \times K}$. In this reverse process, the latent space $\{\boldsymbol{x}_T, \boldsymbol{z}_T, \boldsymbol{z}_{t-1}, ..., \boldsymbol{z}_1\}$ together with the fixed discrete diffusion model $\boldsymbol{\pi}_\theta$ also uniquely define the same stochastic trajectory $\boldsymbol{x}_0, \boldsymbol{x}_1, ..., \boldsymbol{x}_T$. The detailed algorithm is given in Algorithm 2.

### 3.3 ANALYSIS

Here we provide an analysis to quantify the amount of information encoded in latent. Since the inversion involves model forward function call which is difficult to analyze. We describe in the following a simple yet prototypical example of DDPM, where the posterior mean can be computed in closed-form thus allows us to compute the mutual information.

**Remark 3.1.** *Given a simple Gaussian DDPM with $\boldsymbol{x}_0 \sim \mathcal{N}(\boldsymbol{0}, \boldsymbol{I})$, latents $\{\boldsymbol{z}_t\}$ are obtained with DDPM inversion (Huberman-Spiegelglas et al., 2024), then the mutual information between $\boldsymbol{z}_t$ and $\boldsymbol{x}_0$ is:*

$$I(\boldsymbol{z}_t; \boldsymbol{x}_0) = \frac{D}{2} \log\left(\frac{\beta_t^2 \overline{\alpha}_{t-1} + 1 - \overline{\alpha}_{t-1} + \alpha_t(1 - \overline{\alpha}_t)}{1 - \overline{\alpha}_{t-1} + \alpha_t(1 - \overline{\alpha}_t)}\right). \tag{10}$$

The mutual information between $\boldsymbol{z}_t$ and $\boldsymbol{x}_0$ is shown in Figure 3. We observe that the amount of information encoded from $\boldsymbol{x}_0$ into $\boldsymbol{z}_t$ decreases as $t$ increases, motivating us to explore different scheduling strategies for $\lambda$'s (see Supplementary Materials).

## 4 EXPERIMENTS

In this section, we demonstrate the effectiveness of our proposed inversion methods on both image and language diffusion models. Our experiments show that the methods can preserve identity in both vision and language tasks while successfully making the intended changes. The implementation details can be reviewed in Supplementary Materials.

### 4.1 IMAGE DIFFUSION MODEL

For the image diffusion model, we mainly investigate the use of absorbing state discrete model (Austin et al., 2021) including a masked generative model, Paella, and a multinomial diffusion model, VQ-Diffusion. We demonstrate the inversion reconstruction ability and image editing performance in both categories with our Discrete Inversion.

| Method | Metric | | | |
|---|---|---|---|---|
| Inverse+Model | PSNR $\uparrow$ | LPIPS$_{\times 10^3}$ $\downarrow$ | MSE$_{\times 10^4}$ $\downarrow$ | SSIM$_{\times 10^2}$ $\uparrow$ |
| Inpainting+Paella | 10.50 | 565.11 | 1002.09 | 30.13 |
| **Ours+Paella** | **30.91** | **39.81** | **11.07** | **90.22** |
| **Ours[†]+Paella** | Inf | 0.07 | 0.01 | 99.99 |

Table 1: **Inversion Reconstruction performance** † The metric is calculated between the original image and its inverted counterpart. Due to the encoding and decoding steps in the VQ-VAE process, some inaccuracies are introduced by the quantization. The PSNR is inf due to the reconstruction of our method yielding the same image after the VQ-VAE process.

**Dataset.** The Prompt-based Image Editing Benchmark (PIE-Bench) by  (Ju et al., 2023) is a recently introduced dataset designed to evaluate text-to-image (T2I) editing methods. The dataset assesses language-guided image editing in 9 different scenarios with 700 images. The benchmark's detailed annotations and variety of editing tasks were instrumental in thoroughly assessing our method's capabilities, ensuring a fair and consistent comparison with existing approaches.

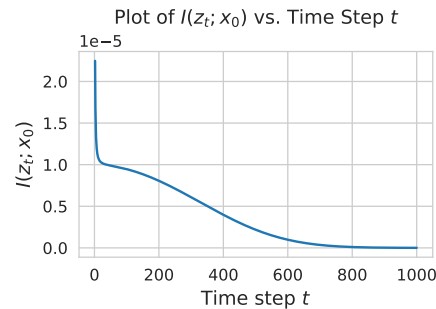

Figure 3: Mutual information between $z_t$ and $x_0$. Computed with a simple DDPM with $x_0 \sim \mathcal{N}(\mathbf{0}, \mathbf{I})$.

#### 4.1.1 INVERSION RECONSTRUCTION

In this section, we evaluate the accuracy of inversion without editing. This is achieved by first inverting the image and then using the recorded latent code to reconstruct the original image.

**Evaluation Metrics** Here, we evaluate the image similarity by PSNR, LPIPS, MSE and SSIM of the original and the generated image under the same prompt with Discrete Inversion and masked generation.

**Quantitative Analysis.** The reconstruction performance of our method, as shown in Table 1, far surpasses the baseline Inpainting + Paella model across all metrics. In the case of masked inpainting, all image tokens are replaced with randomly sampled tokens, meaning the model lacks any prior information about the original image. As a result, the reconstructed image differs significantly from the one being inverted, leading to lower similarity scores. In contrast, our method demonstrates near-perfect reconstruction, as indicated by the metrics, and notably produces an identical image without the errors typically introduced by the VQ-VAE quantization process, as seen in the results marked with †. This highlights the superior accuracy and consistency of our approach in generating high-fidelity reconstructions.

#### 4.1.2 EDITING PERFORMANCE

In this section, we discuss the editing performance of our proposed method. Since there is no discrete diffusion inversion exists, we compare our method with masked generation as indicated in the original paper. In addition to that, we also demonstrate the metric from continuous counterparts.

**Evaluation Metrics.** To demonstrate the effectiveness and efficiency of our proposed inversion method, we employ eight metrics covering three key aspects: structure distance, background preservation, and edit prompt-image consistency, as outlined in Ju et al. (2023). We utilize the structure distance metric proposed by Tumanyan et al. (2023) to measure the structural similarity between the original and generated images. To evaluate how well the background is preserved outside the annotated editing mask, we use Peak Signal-to-Noise Ratio (PSNR), Learned Perceptual Image Patch Similarity (LPIPS) (Zhang et al., 2018), Mean Squared Error (MSE), and Structural Similarity Index Measure (SSIM) (Wang et al., 2004). We also assess the consistency between the edit prompt and the generated image using CLIP (Radford et al., 2021) Similarity Score  (Wu et al., 2021), which is calculated over the whole image and specifically within the regions defined by the editing mask.

| Method | | Structure | CLIP Similarity | |
| --- | --- | --- | --- | --- |
| Inverse | Editing | Distance$_{\times 10^3}$ ↓ | Whole ↑ | Edited ↑ |
| DDIM+SD1.4 | P2P | 69.43* | 25.01* | 22.44* |
| Null-Text + SD1.4 | P2P | 13.44* | 24.75* | 21.86* |
| Negative-Prompt + SD1.4 | P2P | 16.17* | 24.61* | 21.87* |
| DDPM-Inversion + SD1.4 | Prompt | 22.12 | **26.22** | **23.02** |
| ControlNet-InPaint + SD1.5 | Prompt | 65.12 | 25.50 | 22.85 |
| SDEdit ($t_0 = 0.4$) + Paella | Prompt | 30.52 | 23.14 | 20.72 |
| Inpainting + Paella | Prompt | 91.10 | 25.36 | 23.42 |
| **Ours + Paella** | Prompt | **11.34** | 23.79 | 21.23 |
| **Ours + VQ-Diffusion**$^{\dagger}$ | Prompt | 12.70 | 23.85 | 21.02 |

Table 2: **Editing Performance.** We present quantitative results for our proposed method compared to continuous diffusion model (Stable Diffusion v1.4) with DDIM inversion and image inpainting with discrete masked generation model Paella. P2P stands for Prompt-to-Prompt (Hertz et al., 2022), whereas "Prompt" refers to editing solely through the forward edit prompt. Entries marked with asteroids (*) are quoted from Ju et al. (2023). $^{\dagger}$: For VQ-Diffusion, we down-sample the image to $256 \times 256$.

| Method | | Background Preservation | | | |
| --- | --- | --- | --- | --- | --- |
| Inverse | Editing | PSNR ↑ | LPIPS$_{\times 10^3}$ ↓ | MSE$_{\times 10^4}$ ↓ | SSIM$_{\times 10^2}$ ↑ |
| DDIM+SD1.4 | P2P | 17.87 | 208.80 | 219.88 | 71.14 |
| **Ours+Paella** | Prompt | 27.29 | 52.90 | 43.76 | 89.79 |

Table 3: **Background Preservation.** Quantitative comparison of background preservation between our proposed method and DDIM+SD 1.4, achieved by masking the edited region and calculating image similarity with the unedited masked image. The inpainting is served as upper bound since only the masked region are edited and background are not modified.

**Results.** In Table 2, we demonstrate the quantitative result of Discrete Inversion using Paella and VQ-Diffusion compared to continuous diffusion model and also inpainting. Notably, our approach with the Paella model achieves the lowest structure distance 11.34, outperforming all other methods, including the continuous diffusion models. Additionally, while the DDPM Inversion with Stable Diffusion v1.4 shows the highest CLIP similarity scores for both whole and edited regions, our method maintains competitive CLIP similarity with Paella. Given the significant reduction in structure distance, our method offers a superior balance between structural preservation and semantic alignment in edits. Furthermore, when combined with VQ-Diffusion, our method continues to show strong performance. The results in Table 3 clearly demonstrate the superior background preservation capabilities of our method compared to DDIM+SD1.4. All four metrics underscore the structural consistency of our approach in preserving the unedited regions of the image. These results show the effectiveness of our method in maintaining background integrity during editing and provide evidence that information about the original image is instilled into the latent space of Discrete Inversion.

In Figure 4 , we show the editing results for both Paella and VQ-Diffusion using our Discrete Inversion method. Both models successfully modify real images according to the target prompts. In all cases, our results exhibit both high fidelity to the input image and adherence to the target prompt. Additionally, we show the visualization of ControlNet Inpainting and SDEdit results in Figure 11.

## 4.2 Language Diffusion Model

In this section, we evaluate Discrete Inversion on RoBERTa (Liu et al., 2019), a text discrete diffusion model, to generate sentences with opposing sentiments while preserving structural similarities. We begin with two prompts—one with a positive sentiment and another with a negative sentiment. Each prompt contains two sentences: the first sentence indicates the sentiment type and sets the

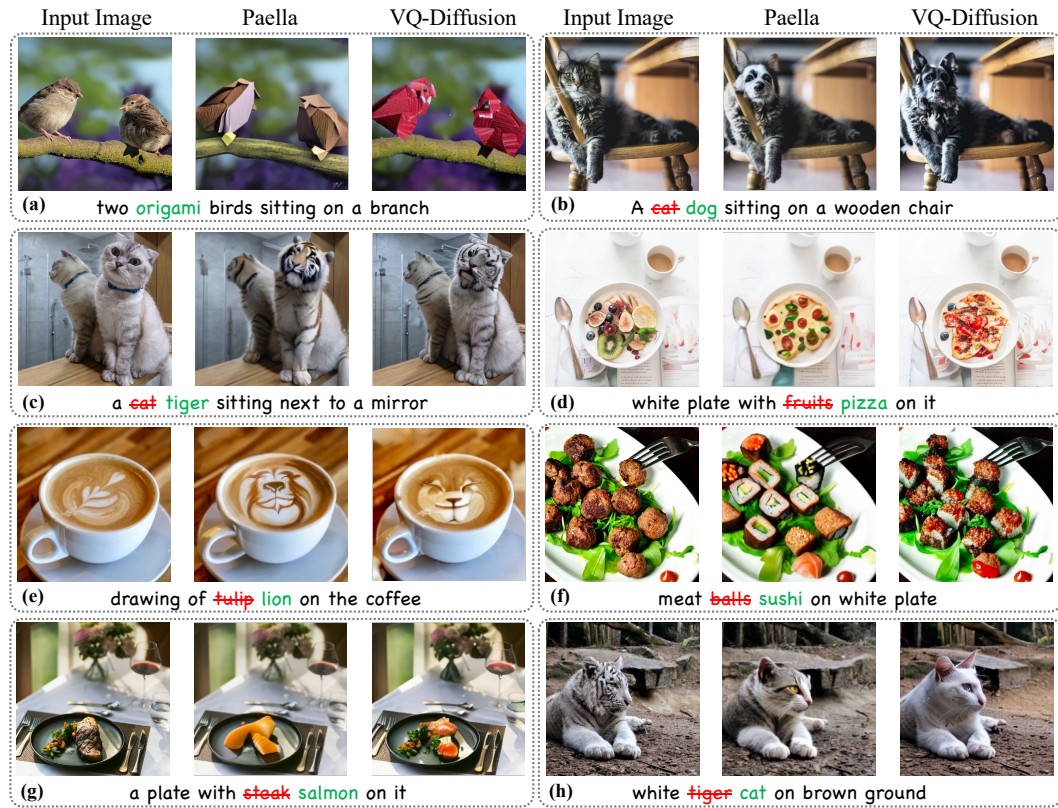

Figure 4: **Visualization of editing results.** Editing results for our method using Paella and VQ-Diffusion are presented, along with their corresponding prompts. The results demonstrate that our method can effectively modify the input image according to the target prompt while preserving the image structure. Editing with masked generative model (Paella (Rampas et al., 2022)) is more stable and easier than with multinomial diffusion models (VQ-Diffusion (Gu et al., 2022)).

contextual background, and the second sentence is the target for inversion and generation. Initially, we invert the second sentence of the negative sentiment prompt using the entire prompt as context, which produces a noised token representation of that sentence. Next, we condition the model on the positive sentiment by concatenating the first sentence of the positive sentiment prompt with the noised token of the inverted negative sentence. This setup guides the model to generate a new second sentence that mirrors the structure of the original negative sentence but expresses a positive sentiment instead. Through this process, we assess the model's capability to invert and generate text that aligns with a specified sentiment while retaining the original sentence's structural elements.

**Inversion Process.** In our experiment, we specifically focus on inverting the second sentence, indicated as red in Table 6, while keeping the first sentence intact (black), as it usually contains essential context. During the reverse process, we aim to reconstruct/edit the second sentence by recovering it from the noised tokens acquired in the inversion phase.

**Dataset Generation.** In order to evaluate the editing performance, we designed and proposed a new dataset called Sentiment Editing. The objective is to edit the sentiment of the sentence while preserving the structure of the sentence and also sticking to the theme of the sentence. Please refer to supplementary materials for the process of generating the dataset and more examples.

### 4.2.1 INVERSION RECONSTRUCTION

Similar to the image generation section, we first demonstrate the inversion and reconstruction capabilities of the proposed methods. This process involves inverting the sentences, followed by using the same prompt to generate the reconstructed version of the second sentence.

**Evaluation Metric.** For reconstruction, we use Hit Rate, which is defined as the proportion of cases where each method generates an identical sentence to the original. In addition, we compute the Semantic Textual Similarity (STS) score by measuring the cosine similarity between the sentence embeddings, using the model proposed by Reimers (2019) *et al*.

**Quantitative Analysis.** Table 4 compares Discrete Inversion with Masked Generation using RoBERTa across two metrics: Accuracy and Semantic Textual Similarity. Our method significantly surpasses Masked Generation in both metrics, demonstrating that our $z_t$ latent space effectively captures the information of the sentence being inverted and facilitates its subsequent reconstruction.

| Method | Metric | |
|---|---|---|
| Inverse+Model | Accuracy$_{\times 10^2}$ ↑ | Textual Similarity$_{\times 10^2}$ ↑ |
| Masked Generation+RoBERTa | 0.0 | 6.57 |
| **Ours+RoBERTa** | **99.74** | **99.90** |

| Method | Metric | |
|---|---|---|
| Inverse+Model | Structure Preservation$_{\times 10^2}$ ↑ | Sentiment Correctness$_{\times 10^2}$ ↑ |
| Masked Generation+RoBERTa | 29.80 | 12.94 |
| **Ours+RoBERTa** | **94.76** | **72.51** |

Table 4: **Text Inversion Reconstruction Performance.** Quantitative comparisons of the text reconstruction performance by Masked Generation and Discrete Inversion method using RoBERTa as the language model.

Table 5: **Text Editing Performance.** Evaluation of the text editing performance between Masked Generation and Discrete Inversion using ChatGPT as a classifier.

### 4.2.2 Sentence Editing

In this section, we evaluate the editing performance of the proposed inversion method on RoBERTa. In Table 6, the sentence shown in black under the negative prompt column is input during the inversion process. The sentence that is being inverted is displayed in red. For editing, the prompt is then substituted with the black sentence on the right, and noise is added at the end for the forward process. The output of the forward process for the noise is presented in blue.

**Evaluation Metric.** For the sentence editing task, we evaluate the generated sentences based on two criteria: (1) structural preservation, which assesses whether the sentence structure is retained, and (2) sentiment correctness, which evaluates whether the sentiment of the edited sentence aligns with the sentiment of the original prompt. Both the structural preservation rate and sentiment correctness rate are calculated using ChatGPT-4 (Achiam et al., 2023) as a classifier. The details of using ChatGPT for evaluation can be reviewed in Supplementary Materials.

**Results.** Table 5 presents a comparative analysis of two text editing methods that both employ RoBERTa, focusing on the effectiveness in terms of Structure Preservation and Sentiment Correctness. Our method significantly outperforms masked generation in both metrics. This difference highlights the superior capability of our inversion method to encode the original structure of the text in the latent space and the flexibility to adjust its sentiment more accurately. In Table 6, we demonstrate both the initial prompt and the edited result. Our approach retains the sentence structure of the negative prompt while modifying its sentiment to a more positive one.

## 5 Conclusion and Discussion

In this paper, we introduced Discrete Inversion, an inversion algorithm for discrete diffusion models, including multinomial diffusion and masked generative models. By leveraging recorded noise sequences and masking patterns during the reverse diffusion process, Discrete Inversion enables accurate reconstruction and flexible editing of discrete data without the need for predefined masks or cross-attention manipulation. Our experiments across multiple models and modalities demonstrate the effectiveness of Discrete Inversion in preserving data fidelity while enhancing editing capabilities. While Discrete Inversion shows promise, we empirically find that editing with multinomial diffusion models may not work as robustly as with masked generative models. Furthermore, it may appear less effective in style transfer tasks, such as transforming an image of a cat into a silver cat statue. Interesting future directions include: (1) developing a more theoretical analysis of mutual information and convergence for continuous and discrete inversion algorithms, (2) extending Discrete Inversion to score distillation sampling (Poole et al.), and (3) exploring the integration of Semantic Guidance (Brack et al., 2023; 2024) within discrete settings.

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

## A  DETAILS ON MULTINOMIAL DIFFUSION MODELS

**Definition of $Q_t$ with mask-and-replace strategy.** Following mask-and-replace strategy as:

$$\boldsymbol{Q}_t = \begin{bmatrix} \alpha_t + \beta_t & \beta_t & \beta_t & \cdots & 0 \\ \beta_t & \alpha_t + \beta_t & \beta_t & \cdots & 0 \\ \beta_t & \beta_t & \alpha_t + \beta_t & \cdots & 0 \\ \vdots & \vdots & \vdots & \ddots & \vdots \\ \gamma_t & \gamma_t & \gamma_t & \cdots & 1 \end{bmatrix}, \tag{11}$$

given $\alpha_t \in [0,1], \beta_t = (1 - \alpha_t - \gamma_t)/K$ and $\gamma_t$ the probability of a token to be replaced with a `[MASK]` token.

**Cumulative transition matrix.** The cumulative transition matrix $\bar{Q}_t$ and $q(x_t|x_0)$ can be computed via closed form:

$$\bar{Q}_t \boldsymbol{v}(x_0) = \bar{\alpha}_t \boldsymbol{v}(x_0) + (\bar{\gamma}_t - \bar{\beta}_t) \boldsymbol{v}(K+1) + \bar{\beta}_t \mathbf{1} \tag{12}$$

where $\bar{\alpha}_t = \prod_{i=1}^{t} \alpha_i, \bar{\gamma}_t = 1 - \prod_{i=1}^{t}(1 - \gamma_i)$, and $\bar{\beta}_t = (1 - \bar{\alpha}_t - \bar{\gamma}_t)/(K+1)$ can be calculated and stored in advance.

## B  ANALYSIS ON MUTUAL INFORMATION

Proof of Remark 3.1.

*Proof.* We assumed that $\boldsymbol{x}_0$ satisfies standard Gaussian distribution $\mathcal{N}(\mathbf{0}, \boldsymbol{I}_D)$. Since

$$\boldsymbol{x}_t = \sqrt{\alpha_t}\boldsymbol{x}_{t-1} + \sqrt{1-\alpha_t}\boldsymbol{\epsilon}_t$$

where both $\boldsymbol{x}_{t-1}$ and $\boldsymbol{\epsilon}_t$ are independent standard Gaussian random variables, $\boldsymbol{x}_t$ is also standard Gaussian, and in each dimension

$$Cov(\boldsymbol{x}_t, \boldsymbol{x}_{t-1}) = \sqrt{\alpha_t},$$

which leads to

$$\hat{\mu}_t(\boldsymbol{x}_t) = \mathbb{E}(\boldsymbol{x}_{t-1}|\boldsymbol{x}_t) = \sqrt{\alpha_t}\boldsymbol{x}_t.$$

Therefore,

$$\begin{aligned} \boldsymbol{z}_t &= \boldsymbol{x}'_{t-1} - \hat{\mu}_t(\boldsymbol{x}_t) \\ &= (\sqrt{\overline{\alpha}_{t-1}}\boldsymbol{x}_0 + \sqrt{1-\overline{\alpha}_{t-1}}\boldsymbol{\epsilon}) - \sqrt{\alpha_t}(\sqrt{\overline{\alpha}_t}\boldsymbol{x}_0 + \sqrt{1-\overline{\alpha}_t}\boldsymbol{\epsilon}') \\ &= \beta_t \cdot \sqrt{\overline{\alpha}_{t-1}}\boldsymbol{x}_0 + \sqrt{1-\overline{\alpha}_{t-1}}\boldsymbol{\epsilon} + \sqrt{\alpha_t(1-\overline{\alpha}_t)}\boldsymbol{\epsilon}'. \end{aligned}$$

Let

$$E = \sqrt{1-\overline{\alpha}_{t-1}}\boldsymbol{\epsilon} + \sqrt{\alpha_t(1-\overline{\alpha}_t)}\boldsymbol{\epsilon}'$$

which is a Gaussian error term independent to $\boldsymbol{x}_0$ with mean 0 and variance $1 - \overline{\alpha}_{t-1} + \alpha_t(1-\overline{\alpha}_t)$. Thus we can calculate the mutual information

$$\begin{aligned} I(\boldsymbol{z}_t; \boldsymbol{x}_0) &= H(\boldsymbol{z}_t) - H(\boldsymbol{z}_t|\boldsymbol{x}_0) \\ &= H(\boldsymbol{z}_t) - H(E) \\ &= \frac{D}{2}\log(2\pi e(\beta_t^2\overline{\alpha}_{t-1} + 1 - \overline{\alpha}_{t-1} + \alpha_t(1-\overline{\alpha}_t))) - \frac{D}{2}\log(2\pi e(1 - \overline{\alpha}_{t-1} + \alpha_t(1-\overline{\alpha}_t))) \\ &= \frac{D}{2}\log(\frac{\beta_t^2\overline{\alpha}_{t-1} + 1 - \overline{\alpha}_{t-1} + \alpha_t(1-\overline{\alpha}_t)}{1 - \overline{\alpha}_{t-1} + \alpha_t(1-\overline{\alpha}_t)}). \end{aligned}$$

□

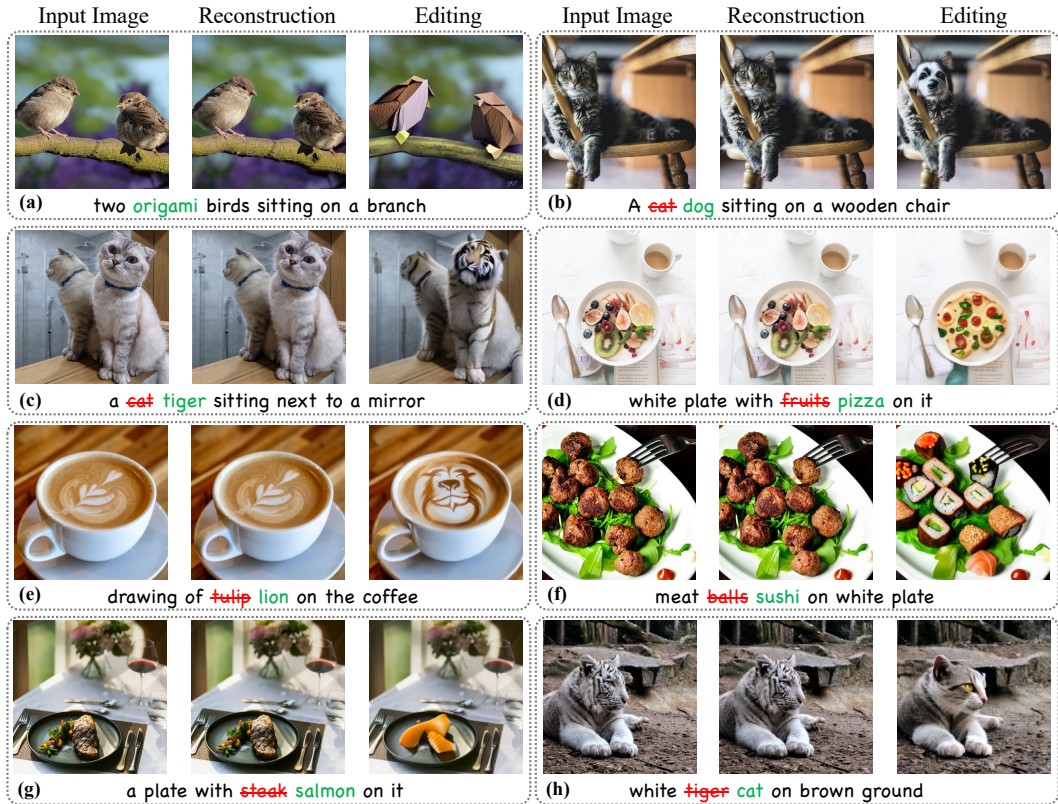

Figure 5: **Reconstruction and editing result with Discrete Inversion and Paella.**

## C    IMPLEMENTATION DETAILS

For all reconstruction task, we employ a $\tau = 1.0$ and $\lambda_1 = 1.0, \lambda_2 = 0.0$ with 32 sampling steps and 26 renoising steps.

The hyper-parameters for Paella editing experiment is CFG= 10.0, $\lambda_1 = 0.7$, $\lambda_2 = 0.3$ and $\tau = 0.9$. The hyper-parameters for VQ-Diffusion in editing is CFG= 5.0, $\lambda_1 = 0.2$, $\lambda_2 = 0.8$.

For sentiment editing task with RoBERTa, we utilize two sets of hyperparameter: $\tau = 0.7$, $\lambda_1 = 0.2$, $\lambda_2 = 0.8$ and $\tau = 0.7$, $\lambda_1 = 0.25$, $\lambda_2 = 0.75$.

All models are implemented in PyTorch 2.0 and inferenced on a single NVIDIA A100 40GB.

## D    ABLATION STUDY

In this section, we analyze the impact of varying hyperparameters $\lambda_1, \lambda_2, \tau$, and CFG scale on the quality of image generation and adherence to textual descriptions, quantified through Structure Distance and CLIP similarity. The hyperparameters play specific roles: $\lambda$ controls the amount of noise introduced in each reverse step, $\tau$ governs the percentage of tokens replaced with random tokens during inversion, and Classifier-Free Guidance (CFG) scales the influence of the text prompt during image synthesis. To limit the search space and simplify the ablation, we choose $\lambda_1 = \lambda$ and $\lambda_2 = 1 - \lambda$ and vary the value of $\lambda$. Evaluation metrics are given in Figure 8.

**Effect of $\lambda_1$ and $\lambda_2$:** With a fixed CFG of 10.0, the graphs indicate that increasing $\lambda$ results in a rise in Structure Distance, suggesting a decline in structural integrity of the images. This increase in noise appears to allow for greater exploration of the generative space at the expense of some loss in image clarity.

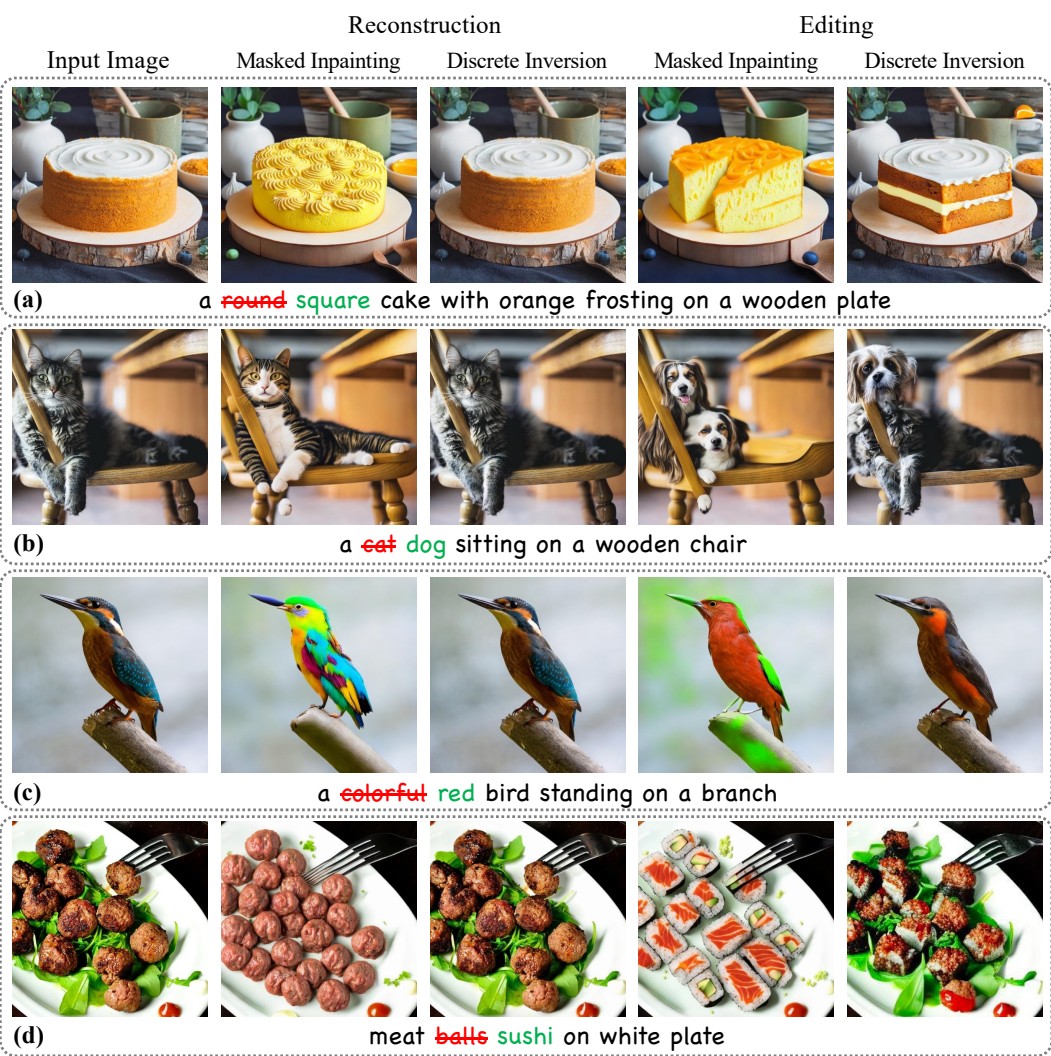

Figure 6: **Reconstruction and editing result with Discrete Inversion and masked inpainting.** Notice that for reconstruction, we use the red prompt, but for editing we use the green prompt.

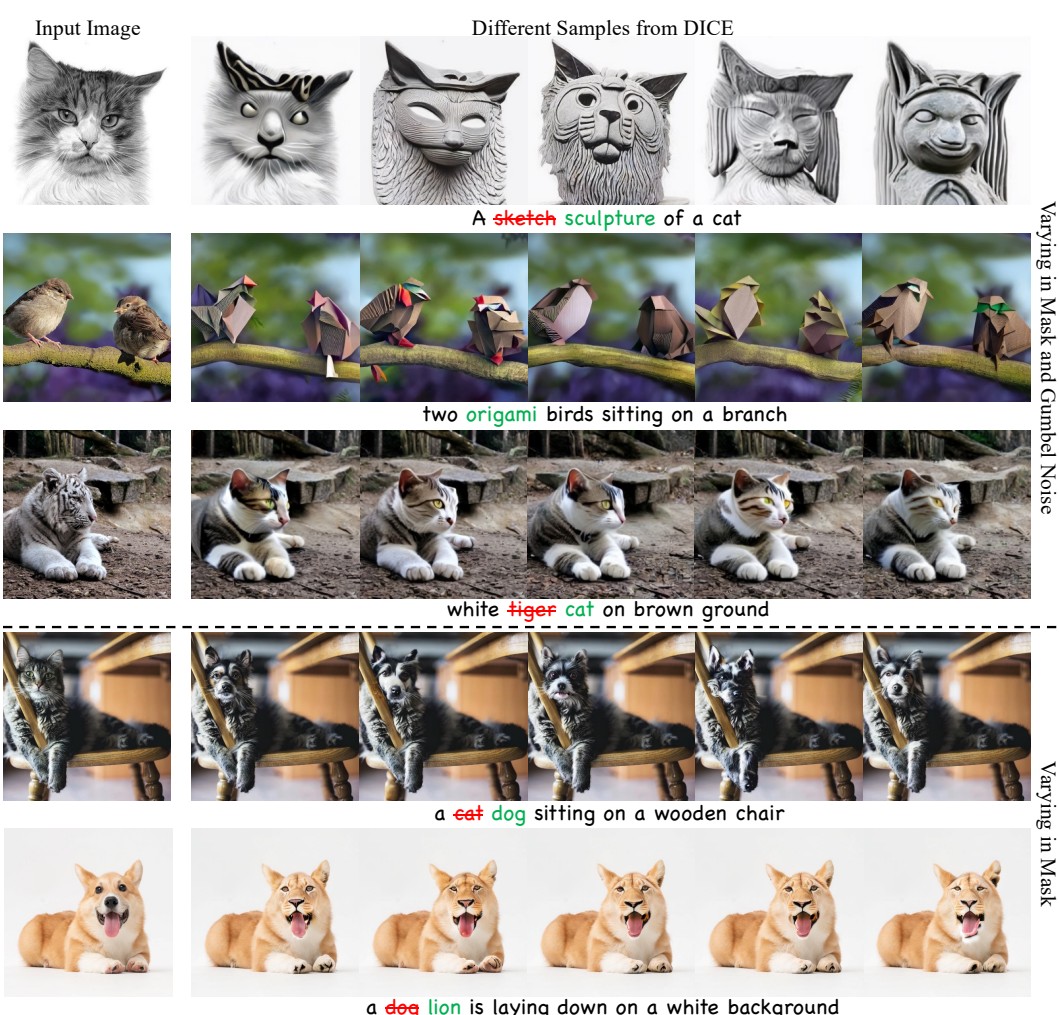

Figure 7: **Image Editing with Diversity.** Due to the stochastic nature of our method, we can generate diverse outputs. The first three rows illustrate variations in both inversion masks and injected Gumbel noise ($\lambda_1 = 0.7$, $\lambda_2 = 0.3$). The last two rows demonstrate variations using only inversion masks ($\lambda_1 = 1$, $\lambda_2 = 0$).

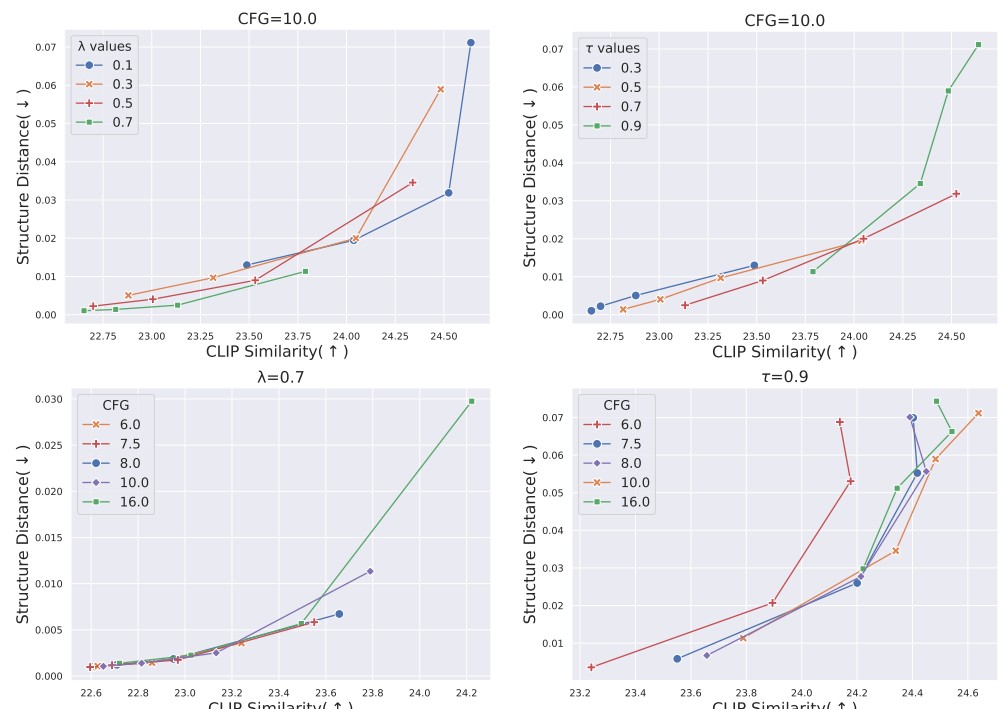

Figure 8: **The effect of hyperparameters $\lambda_1, \lambda_2, \tau$, CFG on the Structure Distance ($\downarrow$) and CLIP similarity ($\uparrow$) with addition function as noise inject function.** In our implementation, to limit the search space, we choose $\lambda_1 = \lambda$ and $\lambda_2 = 1 - \lambda$ for simplicity.

**Effect of $\tau$:** Higher $\tau$ values, particularly at 0.9, show a notable rise in Structure Distance as CLIP similarity increases. This implies that more token replacement can lead to images that align better with the text prompts but may suffer in maintaining structural fidelity, likely due to $\boldsymbol{x}_T$ contains less information of the original image while $\lambda$ injects additional noise during editing phase.

**Effect of CFG Scale:** Varying CFG at a fixed $\lambda$ of 0.7 and $\tau$ of 0.9 reveals that higher CFG values substantially improve Structure Distance, but to an extent (CFG of 10). Beyond this point, further increases in CFG do not yield significant improvements in structural quality, indicating a diminishing return on higher guidance levels. This plateau suggests that while increasing CFG helps in aligning the generated images more closely with the text prompts initially, the benefits in structural integrity and clarity become less visible as CFG values exceed a certain threshold. This finding underscores the need for a balanced approach in setting CFG, where too much guidance may not necessarily lead to better outcomes in terms of image quality and fidelity to the textual description.

**Effect of noise injection function:** We also conducted evaluations using a variance-preserving noise injection function by setting $\lambda_1 = \sqrt{\lambda}$ and $\lambda_2 = \sqrt{1-\lambda}$. The results of these experiments are presented in Figure 9. As for the `max` function, we performed a manual inspection of the visual examples generated with this function. The quality of these examples was noticeably inferior, we therefore omit the corresponding evaluation curves from our analysis.

In conclusion, this ablation study demonstrates that increasing $\lambda$ and $\tau$ can enhance adherence to text prompts through broader explorations in generative spaces, yet this benefit is offset by a decrease in the structural quality of the images. On the other hand, raising CFG values enhances the structural integrity of images to a certain threshold, after which the improvements plateau, indicating a ceiling to the effectiveness of higher CFG settings. This analysis offers empirical guidance for selecting hyperparameters, balancing the trade-offs between text alignment and image quality to optimize image synthesis outcomes.

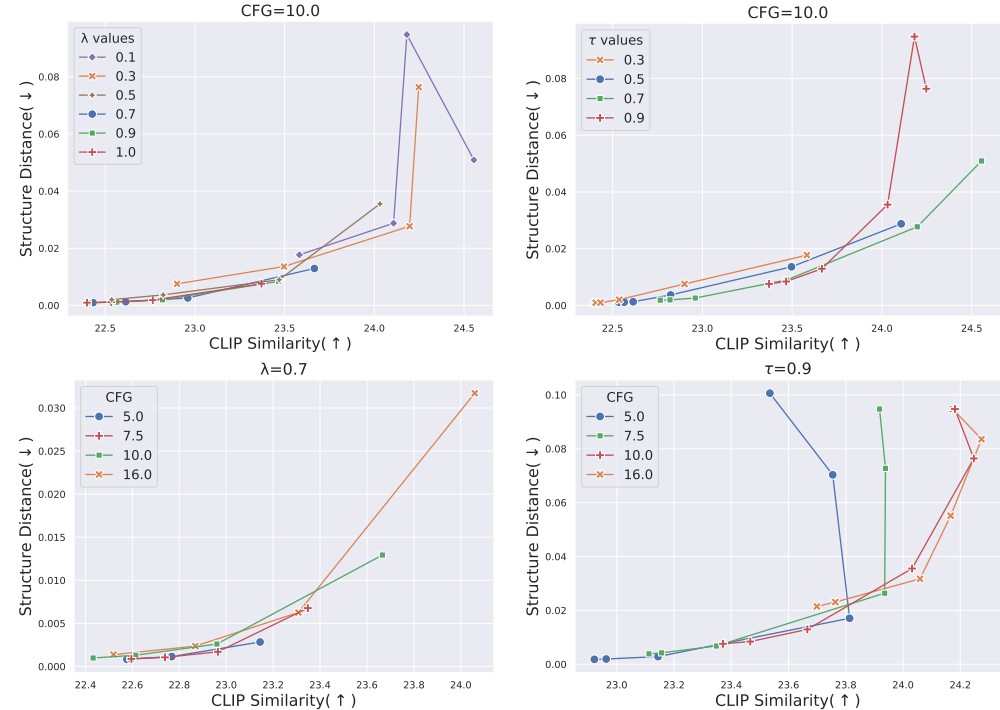

Figure 9: **The effect of hyperparameters $\lambda_1, \lambda_2$ with variance preserving scheme.** We set $\lambda_1 = \sqrt{\lambda}$ and $\lambda_2 = \sqrt{1 - \lambda}$.

## E    ADDITIONAL RESULTS ON IMAGE EDITING

**Reconstruction result with Paella.** In Figure 5 we demonstrates the inversion reconstruction result with Paella using our proposed method.

**Image editing with diversity.** As shown in Figure 7, our method enables diverse image editing results through stochastic variation. The first three rows demonstrate the impact of varying both the inversion masks and the injected Gumbel noise, while the last two rows focus on variations produced by changing only the inversion masks.

## F    ADDITIONAL RESULTS ON TEXT EDITING

**Dataset generation.** To generate the dataset, we utilize ChatGPT-4o with the following prompt:

> **User**
>
> Generate 200 pairs of sentences that contains the same meaning, but one with positive sentiment and one with negative sentiment. For both positive sentiment and negative sentiment, you need to write two sentences with the first part being a hint of the sentiment and the second part being the actual content. The first part for both sentences should be same. write in the format like:
> hint. positive.
> hint. negative.
> Make sure that there are two lines for each pairs. Also, the hint should provide enough context and both positive and negative sentiment should be related to the hint. Do not repeat the hint, also make sure that there is only two sentences in each of the line, one is the hint and the other is about the sentiment.

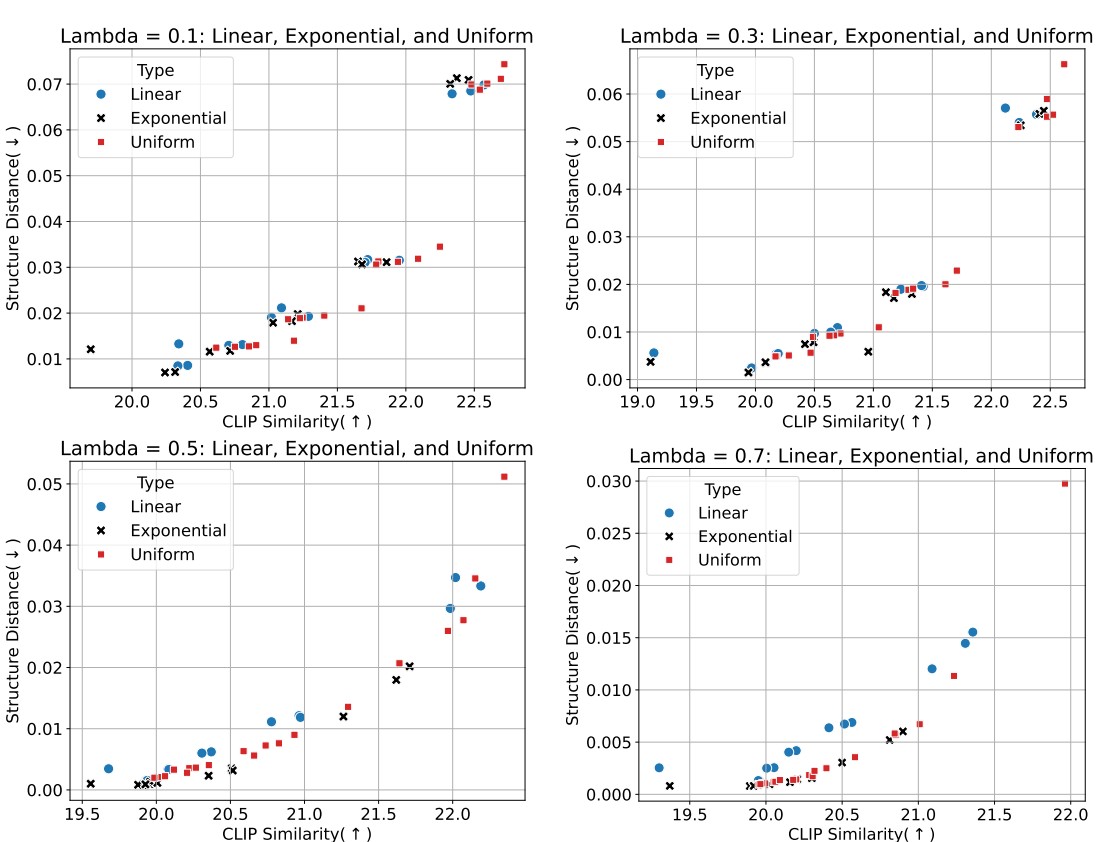

Figure 10: **The effect of different $\lambda$ schedule on the Structure Distance ($\downarrow$) and CLIP similarity ($\uparrow$).** In our implementation, to limit the search space, we choose $\lambda_1 = \lambda$ and $\lambda_2 = 1 - \lambda$ for simplicity.

> **ChatGPT**
>
> 1. Thanks to her efforts. The event was a huge success.
>    Despite her efforts. The event was a complete disaster.
> 2. ...

The sentences is then added with a prefix to indicates the sentiment of the context. Here we demonstrates a subset of our generated dataset:

1. Positive Sentiment: Thanks to her efforts. The event was a huge success.
   Negative Sentiment: Despite her efforts. The event was a complete disaster.

2. Positive Sentiment: This book is definitely interesting. I can't put it down; it's full of surprises.
   Negative Sentiment: This book is definitely interesting. I can't wait to finish it; it's so predictable.

3. Positive Sentiment: The new office space is fantastic. It's spacious and perfect for productivity.
   Negative Sentiment: The new office space is fantastic. It's cramped and lacks proper facilities.

4. Positive Sentiment: Thanks to her efforts. The event was a huge success.
   Negative Sentiment: Despite her efforts. The event was a complete disaster.

5. Positive Sentiment: Regarding the lecture. It was insightful and engaging.
   Negative Sentiment: Regarding the lecture. It was dull and confusing.

6. Positive Sentiment: Despite the initial problems. The project was a success.
   Negative Sentiment: Despite the initial problems. The project ended in failure.

7. Positive Sentiment: Regarding the new app. It's user-friendly and very helpful.
   Negative Sentiment: Regarding the new app. It's complicated and not useful.

8. Positive Sentiment: Reflecting on my environmental initiatives. Implementing changes has reduced my carbon footprint.
   Negative Sentiment: Reflecting on my environmental initiatives. It's challenging to maintain, and progress is slow.

9. Positive Sentiment: The business proposal was well-received. The ideas were innovative, and the presentation was convincing.
   Negative Sentiment: The business proposal was rejected. The ideas were impractical, and the presentation was unconvincing.

10. Positive Sentiment: The training program was highly effective. It boosted skills and confidence, and everyone left motivated.
    Negative Sentiment: The training program was ineffective. It didn't teach much, and most people left feeling unmotivated.

11. ...

## G  EVALUATING THE TEXT EDITING PERFORMANCE

Below, we demonstrate the prompt used for evaluating the editing results:

> **User**
>
> Given three sentences, confirm that the second sentence is roughly the same sentence structure as the first sentence, then confirm that the second sentence has positive sentiment. Output only two numbers with each number indicating whether the corresponding criteria is satisfied. Use 1 for satisfied and 0 for not satisfied. The sentences are given below:
> The event was a complete disaster.
> This event was a fantastic comedy game.

> **ChatGPT**
>
> 1 1

## H    TEXT EDITING RESULTS

| Negative Prompt | Our Edited Results |
|---|---|
| Negative Sentiment: This book is definitely interesting. 
 I can't wait to finish it; it's so predictable. | Positive Sentiment: This book is definitely interesting. 
 I can't wait to see it; it sounds so beautiful. |
| Negative Sentiment: The new office space is fantastic. 
 It's cramped and lacks proper facilities. | Positive Sentiment: The new office space is fantastic. 
 It's spacious and has great facilities. |
| Negative Sentiment: Despite her efforts. 
 The event was a complete disaster. | Positive Sentiment: Thanks to her efforts. 
 This event was a fantastic comedy game. |
| Negative Sentiment: Regarding the lecture. 
 It was dull and confusing. | Positive Sentiment: Regarding the lecture. 
 It was clear and surprising. |
| Negative Sentiment: Despite the initial problems. 
 The project ended in failure. | Positive Sentiment: Despite the initial problems. 
 New project still in progress. |
| Negative Sentiment: Regarding the new app. 
 It's complicated and not useful. | Positive Sentiment: Regarding the new app. 
 It's On and It's Epic. |
| Negative Sentiment: Reflecting on my environmental initiatives. 
 It's challenging to maintain, and progress is slow. | Positive Sentiment: Reflecting on my environmental initiatives. 
 It's easy to understand, and progress is undeniable. |

Table 6: **Editing results of our method with RoBERTa.** The sentences in black are the prompts used for inversion and editing in their respective column. The sentence in red is the one being inverted, and the blue sentence represents the editing result.

## I    ADDITIONAL COMPARISONS

**Additional baselines.** We compare with SDEdit Meng et al. (2021) and ControlNet Zhang et al. (2023a)[1]. Results are shown in Figure 11 and Table 7.

**Noise injection functions.** We compare various noise injection functions, including taking the maximum of Gumbel noise and the recorded noise, as well as the variance-preserving noise injection function.

**Mask schedule functions.** In Figure 13, we present four types of mask scheduling functions: (a, c) concave up and (b, d) concave down. Our results indicate that concave up mask scheduling functions perform better than their concave down counterparts. Quantitative results are shown in Table 8.

**Comparison between inclusive and random masks.** To understand the impact of randomness in the masking schedule, we illustrate masks that are inclusive compared to totally random. Inclusive mask is mask schedule that are increasingly growing, which is used in Paella, compared to randomly sampled masks.

---

[1] We use the ControlNet-InPaint model based on Stable Diffusion v1.5: `https://github.com/mikonvergence/ControlNetInpaint`

SDEdit-0.4  ControlNet-0.5

SDEdit-0.6  ControlNet-1.0

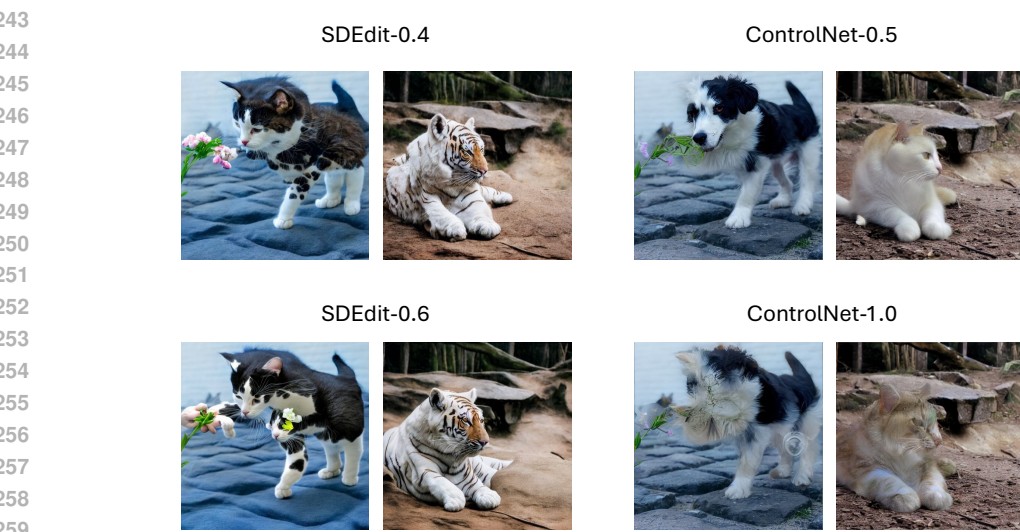

Figure 11: Editing results with SDEdit and ControlNet. For SDEdit we show examples of $t_0 = 0.4, 0.6$. For ControlNet we show examples of conditioning scale of 0.5 and 1.

Maximum

Variance
Preserving

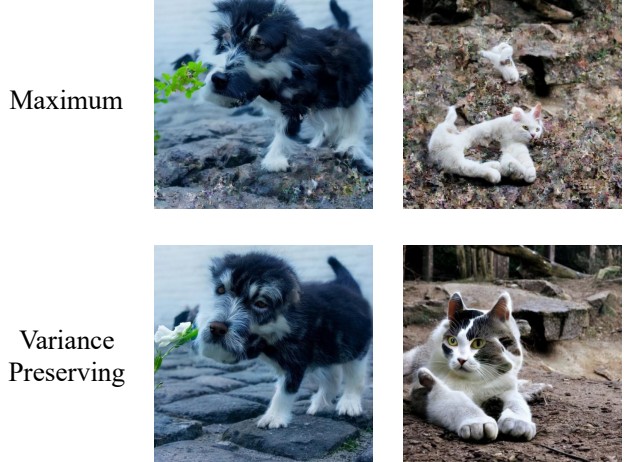

Figure 12: Comparison with different $\lambda$ functions.

| Method | | Structure | CLIP Similarity | |
|---|---|---|---|---|
| Inversion+Model | Editing | Distance$_{\times 10^3}$ ↓ | Whole ↑ | Edited ↑ |
| ControlNet-InPaint (scale=0.5) + SD1.5 | Prompt | 65.12 | 25.50 | 22.85 |
| ControlNet-InPaint (scale=1.0) + SD1.5 | Prompt | 60.87 | 24.35 | 21.40 |
| SDEdit ($t_0 = 0.4$) + Paella | Prompt | 30.52 | 23.14 | 20.72 |
| SDEdit ($t_0 = 0.6$) + Paella | Prompt | 38.62 | 23.22 | 20.86 |
| Inpainting + Paella | Prompt | 91.10 | 25.36 | 23.42 |
| **Ours + Paella** | Prompt | 11.34 | 23.79 | 21.23 |

Table 7: **Additional baselines.** We compare with SDEdit Meng et al. (2021) and ControlNet Zhang et al. (2023a).

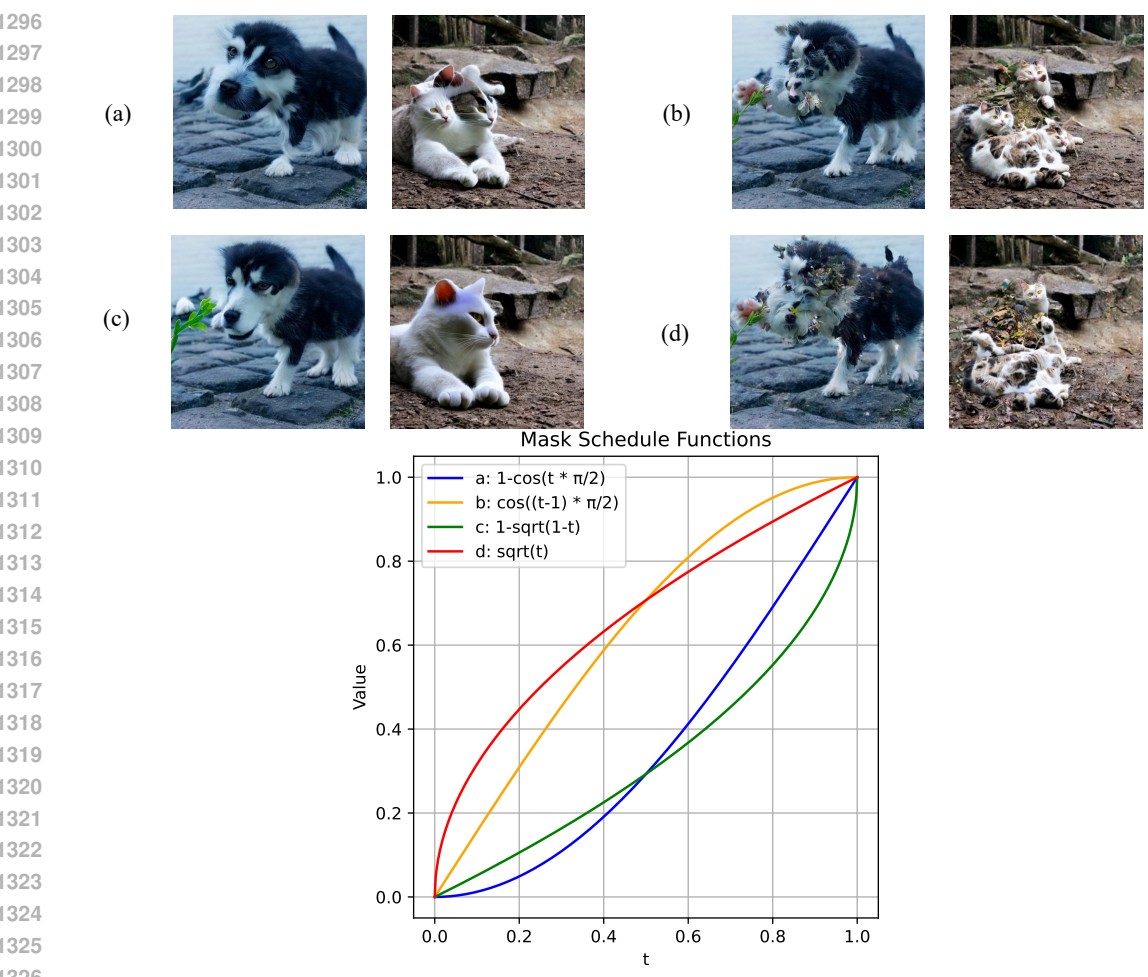

Figure 13: Comparison with different masking schedule. (a): $1 - \cos(t \cdot \pi/2)$, (b): $\cos((t - 1) \cdot \pi/2)$, (c): $1 - \sqrt{1 - t}$, (d): $\sqrt{t}$.

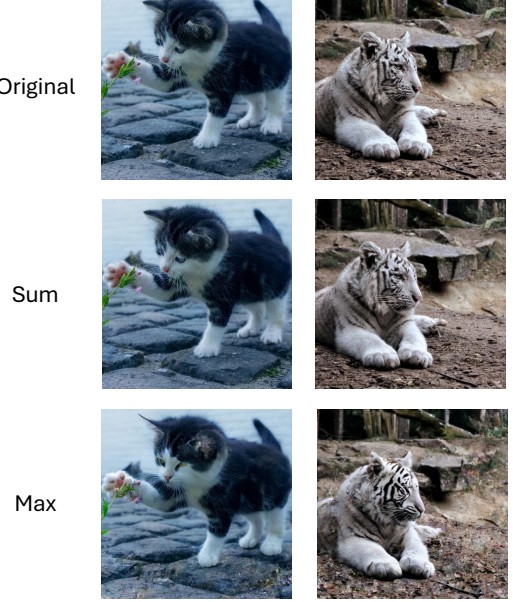

Figure 14: Inversion reconstruction comparison with different noise injection functions.

Inclusive Mask

Random Mask

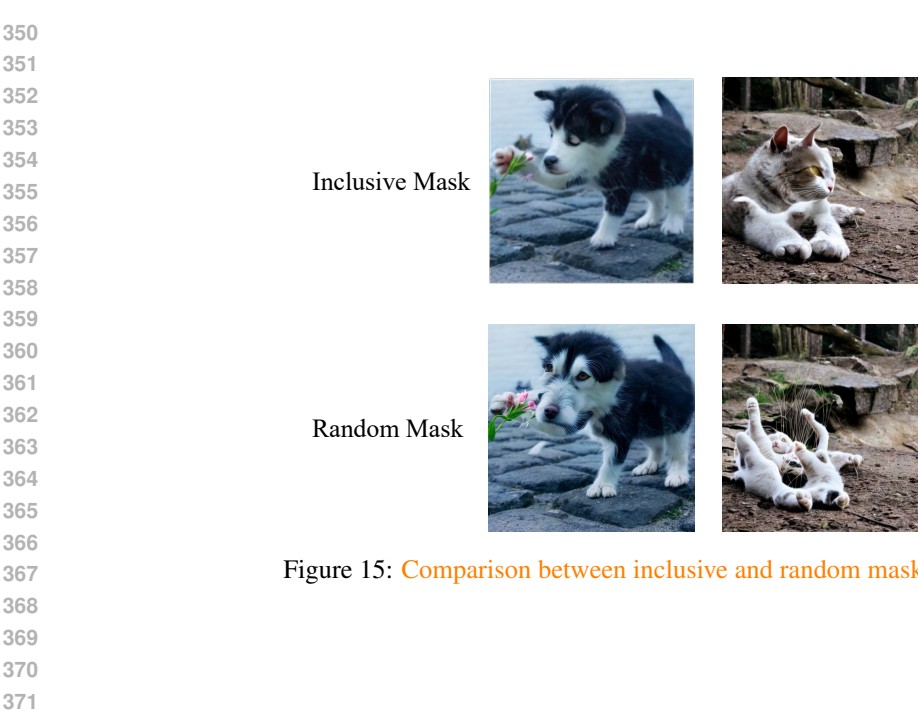

Figure 15: Comparison between inclusive and random masks.

Different Noise between
Inversion and Inference

Different Noise in
Each Sampling
Renoise Step

Different Noise in
Each Renoise Step
for Both Inversion
and Sampling

Ours
(Same Noise in
Each Renoise Step)

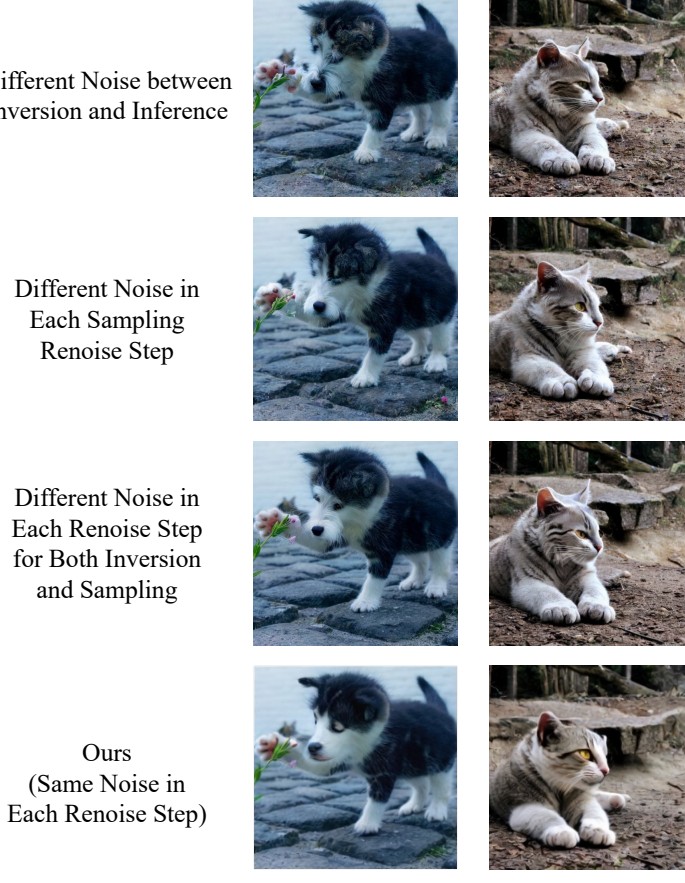

Figure 16: Comparison with using different noise tokens in inversion and inference, using different noise tokens in each step of the renoising step or ours by using the same tokens in both inversion and inference.

| | Structure | CLIP Similarity | |
|---|---|---|---|
| Mask Schedule | Distance$_{\times 10^3}$ ↓ | Whole ↑ | Edited ↑ |
| (a): $1 - \cos(t \cdot \pi/2)$ | 7.54 | 23.48 | 20.96 |
| (b): $\cos((t-1) \cdot \pi/2)$ | 25.39 | 23.56 | 21.24 |
| (c): $1 - \sqrt{1-t}$ | 5.11 | 22.99 | 20.50 |
| (d): $\sqrt{t}$ | 26.35 | 23.59 | 21.36 |
| (e): $t$ | 11.34 | 23.79 | 21.23 |

Table 8: Comparison with different masking schedule. (a): $1 - \cos(t \cdot \pi/2)$, (b): $\cos((t-1) \cdot \pi/2)$, (c): $1 - \sqrt{1-t}$, (d): $\sqrt{t}$.

