# OpenReview forum: "Discrete Inversion: A Controllable Latent Space for Multinomial Diffusion and Masked Generative Models"
_ICLR.cc/2025/Conference — ICLR 2025 Conference Withdrawn Submission_

### Official Review · Reviewer_EPpL · 2024-10-29

**Soundness:** 3
**Presentation:** 2
**Contribution:** 3
**Rating:** 8
**Confidence:** 5

**Summary:**

This paper presents Discrete Inversion, a novel approach for precise inversion in discrete diffusion models.It records noise sequences and masking patterns during forward diffusion.This enables accurate reconstruction and controlled edits without predefined masks or attention map manipulation.The paper demonstrates its effectiveness in both image and text domains.It evaluates the method on models like VQ-Diffusion, Paella, and RoBERTa.Results show high fidelity preservation of original data and flexible editing in discrete spaces.

**Strengths:**

- It seems no one has explored this direction, as far as I know.
- The paper is well-organized across its various sections.

**Weaknesses:**

- The method needs to cache every timestep, so the size is T×D×K. The token number D can be large, e.g., 32×32 in LDM, and K can be quite large too, e.g., 17K. Also, from the last paragraph of the paper on Discrete Flow Matching (Meta), the required timestep is normally larger than its counterpart in continuous space, which further exacerbates the extreme cache burden.
- A theoretical analysis should be provided to show that the inversion indeed makes sense.
- In Line 042, for flow matching, two papers from Michael Albergo are missing.
- Why not directly use categorical sampling instead of using the Gumbel-Softmax trick? maybe because categorical sampling is also inherently using SG trick? the author should make it clear.
- Why is there a minus sign, and what is the motivation in Eq7?
- The algorithm has a drastic issue: it is dependent on the seed. What if we change the seed? It seems the sampling trajectories will not align with the backward process.
- The paper itself feels rushed. I don't understand the details of Alg1 and Alg2. What are lambda1 and lambda2 used for?
- I have no idea why Remark 3.1 is necessary in this draft.
- It's surprising to see the inf result in Table 1, which means no stochastic events happen during the forward and backward processes. Also, I am confused about the difference between the second and third rows in Table 1.
- A basic introduction to RoBERTa is necessary for us to understand your experiment.
- How about if we change the masking schedule? Can it still achieve inversion and editing?
- No code is provided.

**Questions:**

As above

I will further increase my score if my concerns are fully resolved.

---

> ### Author Response · Authors · 2024-11-22
>
> **W1: Cache burden and space efficiency.**
>
> **A1:** Thank you for raising this important point. We acknowledge that the caching requirement could be a potential drawback if cache efficiency is a priority, as it's an inherent limitation of DDPM inversion-based methods. One possible way to mitigate this is by exploring models like Glauber Generative Models \[1\] which turns multiclass into binary classification and thus significantly reduces logit space size. We consider this an exciting avenue for future work.
>
> **W2: Theoretical analysis.**
>
> **A2:** We completely agree with the reviewer on the importance of providing a theoretical foundation. We would like to clarify that analyzing the mutual information in the context of inversion is challenging, as it involves the model's forward call, which is difficult to analyze directly. To address this, we presented a simple yet prototypical example of DDPM where the posterior mean can be computed in closed form. This enabled us to explicitly compute the mutual information, providing valuable insights into the behavior of DDPM Inversion type of approaches.
> That said, we fully acknowledge that a more rigorous theoretical analysis of mutual information and the convergence properties of both continuous and discrete inversion algorithms is necessary. We consider this an important direction for future research and appreciate the reviewer’s suggestion.
>
> **W3: Missing reference.**
>
> **A3:** Thank you for pointing this out, we have referenced the two papers in the revised version.
>
> **W4: Gumbel-Softmax trick.**
>
> **A4:** Yes, that is correct. We have clarified in the revised manuscript.
>
> **W5: Minus sign in Eq 7.**
>
> **A5:** If we rearrange Eq 7 to $y_{t-1} = \hat{y}\_{t-1} + z_t$, we see that $z_t$ is the Gumbel noise or residual that is needed to sample indices $x_{t-1}$ from categorical distribution defined by logits $\hat{y}\_{t-1}$.
>
> **W6: Dependency on random seed.**
> **A6:** We thank the reviewer for raising this interesting perspective. While we respectfully do not see this as a drastic issue, we acknowledge that it is an inherent property of stochastic methods such as non-ODE-based inversion. The algorithm is designed or intended to keep the same seed. To further address this point, we have added additional results in Figure 16 that explore the behavior of our method when the seed is changed under three settings:
> 1. **Row 1:** Different noise maps are used during inversion and inference (editing).
> 2. **Row 2:** A different noise map is used at every renoising step in the sampling process, but the inversion steps only use the same noise map.
> 3. **Row 3:** A different noise map is used at every renoising step in both inversion and sampling steps. In this case, while the same noise map is typically used for all renoising steps in Paella by default, we ensure that the noise maps in corresponding steps during inversion and inference are the same.
>
> Empirical visual results show that our Discrete Inversion is somewhat robust to the variations and randomness in the seed.
>
> **W7: Details on $\lambda\_1$ and $\lambda\_2$.**
>
> **A7:** $\lambda_1$ controls the amount of information injected from the original image, and $\lambda_2$ controls the amount of random noise we want to introduce in inference. Detailed ablations are provided in supplementary section C & D.
>
> **W8: Remark 3.1.**
> **A8:** The analysis in Remark 3.1 is to provide an explicit analysis of the encoded information in the DDPM Inversion type of latents, which is not provided in the original DDPM Inversion paper.
>
> **W9: Clarification of Table 1.**
> **A9:** The "Inf" value in **Table 1** indicates perfect reconstruction, which is a notable property of our method.
> Regarding the difference between the second and third rows in Table 1:
> * The second row measures the distance between the reconstructed image and the **original image**.
> * The third row measures the distance between the reconstructed image and the **VAE-reconstructed image**.
>
> To further clarify, we have added additional visualizations in **Figure 14**, showing examples of reconstruction with different noise injection mechanisms. Please note that when using a max function to merge the recorded noise and sampled Gumbel noise, perfect reconstruction is not guaranteed.

---

> > ### Author Response · Authors · 2024-11-22
> >
> > **W10: Introduction to RoBERTa.**
> >
> > **A10:** Thanks for the valuable suggestion. We have added related discussion of BERT-type models in related work in the revision:
> >
> > > Masked Sequence Modeling has been widely used in representation learning for natural language processing. In models like BERT and RoBERTa, masked tokens ([MASK]) are predicted based on the surrounding context, excelling in text completion and embedding representation learning. BERT-Mouth [1] first interpreted the BERT model as a Markov Random Field and studied its generative perspective. Mask-Predict [2] proposed a similar iterative remask-and-repredict algorithm for machine translation. For image generation, Paella adapts this approach for text-conditional image generation by renoising tokens instead of masking (like in MaskGIT and Muse). These models can be viewed as a special case of discrete diffusion models by introducing an absorbing state. The inference process of these models is typically heuristic and follows a renoise-and-repredict scheme.
> >
> > **W11: Change masking schedule.**
> >
> > **A11:** We have added experiments for various masking schedules (over $t$) in revision **Figure 13** and **Table 8**. Interestingly, in contrast to MaskGIT, which observed that a concave schedule function (e.g., cosine) performs best, we discovered that a convex schedule yields better results in our framework. Additionally, we provide visual examples for a comparison over inclusive mask schedule vs random mask schedule in revision **Figure 15**.
> >
> > | Mask Schedule                      | Structure       | CLIP Similarity     |          |
> > |------------------------------------|-----------------|---------------------|----------|
> > |                                    | Distance × 10³ ↓ | Whole ↑            | Edited ↑ |
> > | (a): $\(1 - \cos(t \cdot \pi / 2)\)$ | 7.54            | 23.48              | 20.96    |
> > | (b): $\(\cos((t - 1) \cdot \pi / 2)\)$ | 25.39           | 23.56              | 21.24    |
> > | (c): $\(1 - \sqrt{1 - t}\)$          | 5.11            | 22.99              | 20.50    |
> > | (d): $\(\sqrt{t}\)$                  | 26.35           | 23.59              | 21.36    |
> > | (e): $\(t\)$                         | 11.34           | 23.79              | 21.23    |
> >
> > **W12: Code release.**
> >
> > **A12:** We will release code upon acceptance.

---

> > ### Comment · Reviewer_EPpL · 2024-11-22
> > **for w1, how is the space efficiency? do you have some number?**
> >
> > as title

---

> ### Author Response · Authors · 2024-11-22
> **Space efficiency**
>
> We appreciate the reviewer’s prompt response.
>
> To store the \$z_t\$’s for Paella for an image of 256×256 (token map 64×64), each \$z_t\$ is stored as torch.float16, with the shape of \$[1, 8192, 64, 64]\$, so storing each \$z_t\$ needs 67.11MB in GPU memory. We need to store a total of 31 maps, which is about 2080MB (~2GB). However, this tensor does not require gradients and can be loaded on demand. Two possible approaches to improve efficiency:
>
> 1.	Compression: Since the logit map does not require very high precision, we can use further quantization to reduce cache size or employ sparse representations.
> 2.	Alternative Models: Using models like Glauber Generative Models, which convert multiclass classification into binary classification, could achieve up to an 8,000× memory saving.
>
> Overall, the entire process fits within 16 GB of VRAM, making it feasible to run on consumer-grade GPUs.

---

> > ### Author Response · Authors · 2024-11-25
> > **Sincerely Awaiting Your New Feedback (Within 2 Days Left)**
> >
> > Dear Reviewer EPpl,
> >
> > Thank you once again for your time and effort in reviewing our work. With only `<2 days` remaining, we kindly request your feedback on our new response. If any part of our explanation is unclear, please let us know. We would greatly appreciate it if you could confirm whether your concerns have been addressed. If the issues are resolved, we would be grateful if you could consider reevaluating the work. Should you need any further clarification, we are happy to provide it promptly before the discussion deadline.
> >
> > Thanks.
> >
> > Authors

---

> > > ### Comment · Reviewer_EPpL · 2024-11-26
> > > **thank you**
> > >
> > > I have raised my score

---

> > > > ### Author Response · Authors · 2024-11-26
> > > > **Thank you for your feedback**
> > > >
> > > > Thank you for your thoughtful feedback and for taking the time to review our work. We truly appreciate your constructive comments, which have been very helpful in identifying areas for improvement.

---

### Official Review · Reviewer_qRGw · 2024-11-03

**Soundness:** 3
**Presentation:** 4
**Contribution:** 3
**Rating:** 6
**Confidence:** 4

**Summary:**

This paper proposes an inversion approach, Discrete Inversion, for discrete diffusion models. By offering a faithful inversion method that enables edits for discretized models, they enable image and text editing without the need of manipulating the cross-attention maps, unlike many of the editing methods operating on continuous diffusion models. Instead of directly manipulating such features, Discrete Inversion proposes saving intermediate latents and masks during inversion and leveraging them during sampling. With the proposed inversion algorithm, authors empirically prove that accurate construction is guaranteed and address the editability of the inverted samples with additional hyperparameters, where each are ablated throughout the experiments provided.

Following a detailed description of the proposed algorithm, authors demonstrate the effectiveness of their method on both text and image modalities both in terms of editing and reconstruction performance. Over the experiments provided, authors present comparisons with continuous diffusion models for editing task and inpainting coupled with discrete diffusion models for image domain. For textual editing results, the experiments compare masked generation with the proposed approach by both assessing the model by RoBERTa itself (Accuracy and Textual Similarity) and by using ChatGPT as a classifier.

**Strengths:**

- Authors introduce Discrete Inversion, which serves as the first inversion method for masked diffusion models, which enables applications such as image editing with a pipeline different than inpainting based editing.
- The theoretical foundations of the proposed algorithm is expressed clearly.
- Editing and reconstruction performance of the proposed method is demonstrated with sufficient metrics that shows the effectiveness of the methods.
- In addition to the proposed inversion algorithm, authors propose a dataset for semantic editing on text which can be leveraged by future studies as a standardized benchmark.

**Weaknesses:**

- The authors provide some claims that are now completely correct, which can be considered as over-generalization. Between lines 154-157, authors generalize the editing methods for diffusion models into two classes which are prompt-based editing with DDPM-based methods and cross-attention manipulation based methods with DDIM-based methods. Such approaches are not the only ways to perform semantic editing on images, whereas techniques like semantic guidance [1, 2] also exists which does not interfere with the main generation prompt and the cross-attention maps. Even though such approaches may not be suitable for comparisons, such claims should be corrected as it can be falsifying. If there is a specific reason that the authors only do consider cross-attention map manipulation based edits for continuous diffusion models, it should be explained clearly.
- $Cat(x;\pi)$ operator should be clarified before its first usage in line 181, as Eq. 1. This would improve readability of the proposed algorithm.
- For an accurate comparison between the approaches based on continuous diffusion models and masked generative models, the authors should expand the comparisons on Table 2 with mask based ControlNet[3]. This baseline can also simulate the effect of preserving unmasked regions (unedited regions). Such a comparison would be more reliable as it will operate on the same masked conditions, where continuous diffusion models by themselves do not have such restrictions in the way masked diffusion models have.

[1] SEGA: Instructing Text-to-Image Models using Semantic Guidance, Brack et. al., https://arxiv.org/abs/2301.12247, NeurIPS 2023

[2] LEDITS++: Limitless Image Editing using Text-to-Image Models, Brack. et. al. https://arxiv.org/abs/2311.16711, CVPR 2024

[3] Adding Conditional Control to Text-to-Image Diffusion Models, Zhang et. al. https://arxiv.org/abs/2302.05543, ICCV 2023

**Questions:**

- Authors should describe why they consider RoBERTa as a diffusion model. RoBERTa is not natively a diffusion model but a masked generative model. However, the model is described as a text discrete diffusion model in line 377. Please explain the relationship and the connection between the masked generative modeling and diffusion models explicitly, if you made any related assumption.
- For clarity, authors should consider including a brief definition of the operator $Cat(x;\pi)$
- Additionally, authors are encouraged to reflect on the weaknesses mentioned both in terms of textual clarity and experiments that can result in a more robust baseline.
- Is there any reason why ControlNet is not included as a part of the comparisons? If there is a specific reason, please explain.

---

> ### Author Response · Authors · 2024-11-22
>
> **W1: Related work discussion.**
> **A1:** We thank the reviewer for the valuable suggestion, and we have updated this in the revised submission. We agree that exploring the integration of SEGA-like approaches in discrete settings could be an interesting direction for future work.
>
> **W2 & Q2: Operator definition.**
> **A2:** We thank the reviewer for pointing this out, and we have addressed this issue in the revised submission.
>
> **W3 & Q4: Comparison with ControlNet.**
> **A3:** We thank the reviewer for this valuable suggestion and have added comparisons with ControlNet in **Table 7** and **Figure 11** in the revised submission. In the original manuscript, we did not include ControlNet for the following reasons:
> 1. **Base model differences:** ControlNet is built on a continuous diffusion base model (e.g., Stable Diffusion), while our method focuses on discrete diffusion models (e.g., Paella). These fundamental differences make direct comparisons less straightforward.
> 2. **Limitations of ControlNet:**
>    While ControlNet offers significant flexibility and controllability for introducing additional conditioning inputs (e.g., skeletons, edges, or depth), it has certain limitations. For instance:
>    * Users often need to manually specify the subject or object to edit, and the success depends on the availability of corresponding detection models (e.g., there might not be a detection model for "dog skeletons").
>    * It requires dataset collection and pretraining if no pre-trained ControlNet exists for a specific use case.
>    * It cannot guarantee perfect reconstruction. For example, while it may accurately control the pose of a dog, the generated dog might not resemble the original or specific target (e.g., a cat transformed into a similar-looking dog).
> 3. **Scope and intent of our paper:**
>    Our primary objective is to introduce a complementary perspective for discrete diffusion models, expanding the existing toolbox. Importantly, our method and ControlNet are not mutually exclusive but rather complementary. If a ControlNet were developed for discrete models like Paella or VQ-Diffusion, our approach could be seamlessly integrated and used alongside it.
>
> We believe that these clarifications, along with the newly added comparisons, provide a more comprehensive understanding of our method in relation to ControlNet.
>
> | Method                            | Editing | Structure |             | CLIP Similarity |          |
> |-----------------------------------|---------|-----------|-------------|-----------------|----------|
> |                                   |         | Distance × 10³ ↓ | Whole ↑    | Edited ↑       |
> | ControlNet-InPaint (scale=0.5) + SD1.5 | Prompt  | 65.12     | 25.50       | 22.85          |
> | ControlNet-InPaint (scale=1.0) + SD1.5 | Prompt  | 60.87     | 24.35       | 21.40          |
> | SDEdit (t₀ = 0.4) + Paella       | Prompt  | 30.52     | 23.14       | 20.72          |
> | SDEdit (t₀ = 0.6) + Paella       | Prompt  | 38.62     | 23.22       | 20.86          |
> | Inpainting + Paella              | Prompt  | 91.10     | 25.36       | 23.42          |
> | **Ours + Paella**                | Prompt  | 11.34     | 23.79       | 21.23          |

---

> > ### Author Response · Authors · 2024-11-22
> >
> > Q1: Discussion about RoBERTa.
> >
> > Ans: We thank the reviewer for this valuable suggestion and have added related discussion of BERT-type models in related work in the revision:
> >
> > > Masked Sequence Modeling has been widely used in representation learning for natural language processing. In models like BERT and RoBERTa, masked tokens (\[MASK\]) are predicted based on the surrounding context, excelling in text completion and embedding representation learning. BERT-Mouth \[1\] first interpreted the BERT model as a Markov Random Field and studied its generative perspective. Mask-Predict \[2\] proposed a similar iterative remask-and-repredict algorithm for machine translation. For image generation, Paella adapts this approach for text-conditional image generation by renoising tokens instead of masking (like in MaskGIT and Muse). These models can be viewed as a special case of discrete diffusion models by introducing an *absorbing state*. The inference process of these models is typically heuristic and follows a renoise-and-repredict scheme.
> >
> > Q3: Discussion of limitations.
> >
> > Ans: We thank the reviewer for this valuable suggestion and have added discussions and limitations in conclusion section:
> >
> > > While Discrete Inversion shows promise, we empirically find that editing with multinomial diffusion models may not work as robustly as with masked generative models. Furthermore, it may appear less effective in style transfer tasks, such as transforming an image of a cat into a silver cat statue. Interesting future directions include: (1) developing a more theoretical analysis of mutual information and convergence for continuous and discrete inversion algorithms, (2) extending Discrete Inversion to score distillation sampling \[3\], and (3) exploring the integration of Semantic Guidance \[4,5\] within discrete settings.
> >
> > \[1\] Wang, Alex, and Kyunghyun Cho. "BERT has a mouth, and it must speak: BERT as a Markov random field language model." *arXiv preprint arXiv:1902.04094* (2019).
> >
> > \[2\] Ghazvininejad, Marjan, Omer Levy, Yinhan Liu, and Luke Zettlemoyer. "Mask-predict: Parallel decoding of conditional masked language models." *arXiv preprint arXiv:1904.09324* (2019).
> >
> > \[3\] Poole, Ben, Ajay Jain, Jonathan T. Barron, and Ben Mildenhall. "DreamFusion: Text-to-3D using 2D Diffusion." In *The Eleventh International Conference on Learning Representations*.
> >
> > \[4\] SEGA: Instructing Text-to-Image Models using Semantic Guidance, Brack et. al., [https://arxiv.org/abs/2301.12247](https://arxiv.org/abs/2301.12247), NeurIPS 2023
> >
> > \[5\] LEDITS++: Limitless Image Editing using Text-to-Image Models, Brack. et. al. [https://arxiv.org/abs/2311.16711](https://arxiv.org/abs/2311.16711), CVPR 2024

---

> > > ### Comment · Reviewer_qRGw · 2024-11-25
> > >
> > > Thanks to the authors for addressing my concerns. I raise my score to 6.

---

> > > > ### Author Response · Authors · 2024-11-25
> > > > **Thank you for your feedback**
> > > >
> > > > We sincerely thank the reviewer qRGw for your thoughtful feedback and for taking the time to reconsider their evaluation. We greatly appreciate your comments, which have helped us improve the clarity and quality of our work.

---

### Official Review · Reviewer_Vpe1 · 2024-11-04

**Soundness:** 2
**Presentation:** 2
**Contribution:** 1
**Rating:** 3
**Confidence:** 5

**Summary:**

This paper provides controlled editing of discrete diffusion models by discrete inversion.

Discrete inversion records transition sequences and masking patterns during the forward diffusion process.

For editing, the proposed method uses partial inversion and re-generation.

Experiments evaluate the proposed method on discrete image diffusion models and language diffusion models.

**Strengths:**

1. The problems and the solutions are clear and sound.
    1. inversion regarding discrete diffusion models -> DDPM inversion
    2. editing -> SDEdit-like partial inversion and generation
2. Evaluations are well-structured.

**Weaknesses:**

1. Recording-based approach is trivially adopted from DDPM inversion to discrete diffusion models: recording transitions and masks instead of noises. Please describe non-trivial differences due to problem settings such as discrete property.
2. Solving editing with partial inversion and re-generation is trivially adopted from SDEdit. Please describe non-trivial differences due to problem settings such as discrete property.
3. Analysis of the mutual information between z_t and x_0 is trivial: it decreases along t. Please describe non-trivial aspects of this tendency.
4. The choice of the competitor is not sound. Why should the masked inpainting be the competitor? In addition, DDIM + SD1.4 + P2P is outdated (2022). DDPM inversion (2024) should suffice.
5. Introduction wastes too much space on continuous diffusion models. It should better describe discrete diffusion models and masked generative models.
6. Section 3.2 contains too much details of previous methods.

**Questions:**

Questions are merged in the weaknesses because they are closely connected.

---

> ### Author Response · Authors · 2024-11-22
>
> **W1: Highlight key differences from DDPM Inversion.**
> **A1:** While extending DDPM Inversion to multinomial diffusion is more straightforward, our main contribution is discrete inversion for masked generative models. The key differences between our approach and DDPM inversion are as follows:
>
> 1. **Inference process differences.** In DDPM Inversion, the inference process involves "fitting" one inference step by recording the Gaussian noise required to sample a given $x\_{t-1}$ from the Gaussian posterior distribution given $x\_t$. This is very different from masked generative models, where $x\_{t-1}$ is NOT sampled from a posterior given $x\_t$. instead, $x\_t$ is obtained from sampled $\\hat{x}\_{0|t}$ by renoising. Thus our discrete inversion is to find the Gumbel noise needed in order to sample the logit of $x\_0$ given $x\_t$. So the method is NOT straightforwardly adaptable from continuous to discrete.
> 2. **Noise type difference.** Our method operates in a discrete domain using Gumbel distributions, as opposed to the continuous domain with Gaussian distributions used in DDPM Inversion. This necessitates addressing challenges to discrete generative processes, such as determining appropriate noise injection mechanisms and studying a suitable noise injection function.
> 3. Another challenge lies in handling the complexities of operating in the **logit space**. Converting indices to logits accurately is itself a non-trivial problem that we address in our work. For Paella model, we use the model conditioned on $t=0$ as the conversion function, i.e., $\\mathcal{D}(\\cdot, t=0)$. We empirically verified the accuracy of this approach in index space and the mean squared error in logit space. For VQ-Diffusion, we use log onehot as Multinomial Diffusion paper implements everything in logit space and this function is compatible with their implementation.
> 4. We also conducted ablation studies on several key **decision choices** in our framework. These include the mask schedule, which differs from DDPM inversion. Unlike DDPM inversion, where an independent q-sample is always used (corresponding roughly to random masking). Visual comparison results are shown in revision **Figure 15**. We also tested various masking schedules over $t$, with results shown in revision **Figure 13** and **Table 8**. Additionally, we investigated the impact of initial noise and the renoising process on the overall performance (see revision **Figure 16**). These experiments provide valuable insights into the roles of these components. For instance, in contrast to MaskGIT, which observed that a concave schedule function (e.g., cosine) performs best, we discovered that a convex schedule yields better results in our framework.
> 5. **Noise injection.** We introduce a mechanism to control the amount of information injected in editing inference, i.e., down weighting the recorded residual (or latent) and add random Gumbel noise. We also discussed and ablated various injection functions (linear, **max**, variance preserving). Without this mechanism, simply adding back the recorded residual with a scale of 1 often results in the model failing to edit and instead producing a reconstruction.
> 6. We highlight our **key contributions**:
>    1. We introduce the first inversion algorithm for discrete diffusion models including both masked generative models and multinomial diffusion.
>    2. We experimented on both image and text modalities. For text, we showed an interesting application to turn a BERT model into a text editing model.
>    3. We introduce a new dataset for text editing evaluation.

---

> > ### Author Response · Authors · 2024-11-22
> >
> > **W2: Highlight key differences from SDEdit.**
> > **A2:** We would like to clarify that we do NOT claim solving editing through partial inversion and re-generation as a novel contribution of our method. Instead, we incorporate this well-known technique as a hyperparameter to enhance the controllability of the editing process. We have added SDEdit as an additional baseline and report results in revision **Figure 11** and **Table 7**.
> > | Method                            | Editing | Structure |             | CLIP Similarity |          |
> > |-----------------------------------|---------|-----------|-------------|-----------------|----------|
> > |                                   |         | Distance × 10³ ↓ | Whole ↑    | Edited ↑       |
> > | SDEdit (t₀ = 0.4) + Paella       | Prompt  | 30.52     | 23.14       | 20.72          |
> > | SDEdit (t₀ = 0.6) + Paella       | Prompt  | 38.62     | 23.22       | 20.86          |
> > | Inpainting + Paella              | Prompt  | 91.10     | 25.36       | 23.42          |
> > | **Ours + Paella**                | Prompt  | 11.34     | 23.79       | 21.23          |
> >
> > **W3: Analysis of mutual information.**
> > **A3:** We respectfully disagree with the assertion that a decreasing tendency in mutual information makes this analysis "trivial." This quantitative analysis was not included in the DDPM Inversion paper, and we believe it provides a more grounded understanding of mutual information in discrete diffusion models. This understanding, in turn, motivated our exploration of scheduling strategies to decay $\\lambda$.
> >
> > Furthermore, analyzing the mutual information is inherently challenging because the inversion process involves the model's forward function, which is difficult to analyze directly. To address this, we presented a simple yet prototypical example of DDPM where the posterior mean can be computed in closed form. This allows us to compute the mutual information explicitly, offering valuable insights into the behavior of discrete diffusion models that are not otherwise apparent.
> >
> > **W4: Choice of baselines.**
> > **A4:** We would like to clarify that the DDIM \+ SD1.4 \+ P2P results were quoted directly from the Direct Inversion paper \[1\] for readers' reference. Additionally, we included DDPM Inversion on PIE-bench as an additional baseline in our experiments. The only direct comparable baseline is masked inpainting and SDEdit (added in revision, see **Figure 11** and **Table 7**) with the same base model (i.e. Paella). A direct comparison between our method and other baselines is less fair and meaningful since the base models are completely different, and were trained on different datasets.
> >
> > **W5: Introduction writing.**
> > **A5:** Thank you for the suggestion. In the revision, we have restructured the introduction section to allocate more focus to discrete diffusion models and masked generative models.
> >
> > **W6: Method section writing.**
> > **A6:** In Section 3.2, only one paragraph on non-ODE-based inversion is related to prior work. The intention was to provide necessary context and a natural introduction to our concept, particularly for readers who may not be familiar with related methods like DDPM Inversion.
> > That said, we appreciate the reviewer’s suggestion and have restructured the method section in the revision to more clearly highlight the key differences and contributions specific to discrete models.
> >
> > \[1\] Ju, Xuan, Ailing Zeng, Yuxuan Bian, Shaoteng Liu, and Qiang Xu. "Direct inversion: Boosting diffusion-based editing with 3 lines of code." *arXiv preprint arXiv:2310.01506* (2023).

---

> > > ### Comment · Reviewer_Vpe1 · 2024-11-24
> > >
> > > I thank the authors for the rebuttal.
> > >
> > > W1) For me, it is straightforward to adapt DDPM inversion by simply replacing Gaussian noise with Gumbel noise because they define the forward diffusion process in continuous and discrete cases. The obvious techniques simply follow this change.
> > >
> > > W2) OK
> > >
> > > W3) Sorry for my mistake. I meant to say "apparent", rather than "trivial". I think it is apparent because the larger noise the more removes the original information. Can authors explain why it is not apparent?
> > >
> > > W4) OK
> > >
> > > W5) Not satisfactory but I appreciate the effort.
> > >
> > > W6) OK

---

> > > > ### Author Response · Authors · 2024-11-25
> > > > **Response to W3**
> > > >
> > > > A3: We thank the reviewer for the clarification and for keeping the discussion open. While we acknowledge that "apparent" is a subjective judgment and not something we can argue against, we kindly reiterate that this quantitative analysis was not included in the DDPM Inversion paper. We also note that we have not claimed this analysis as a main contribution; so it was presented as a remark not sugar-coated as a theorem. A rigorous theoretical analysis remains challenging since computing \$z_t\$  involves model forward calls and distribution of \$x_0\$ can be arbitrarily complicated. Our current analysis is based on simplified assumptions, serving as an initial step toward a deeper theoretical understanding of the problem. We hope this addresses the reviewer’s concern.

---

> ### Author Response · Authors · 2024-11-28
>
> A1:
>
> Thank you for your feedback.
>
> It is indeed fundamental to think of gumbel noise for discrete model in replace of Gaussian noise for continuous model.  While our algorithm 2 can be considered as the extension of DDPM-inversion for discrete space, our **algorithm 1** is **totally different** from DDPM-inversion. As shown in the line 7 of algorithm 1, our gumbel noise $z_t$ is obtained via the difference between ground truth \$y_0\$ and predicted $ \hat{y}\_{0\|t}$ , which is different with the line 7 of algorithm 2, the gumbel noise is obtained from the difference between $y\_{t-1}$ and $\hat{y}\_{t-1|t}$, which logit at time $t-1$ is obtained from posterior sampling. Below here, we will reason the design of 2 proposed inverse algorithms 1 and 2 which is heavily based on the training procedure of generative model.
>
> **1: Paella, MaskGit and Bert training algorithm**
>
> Given data $x\_{0}$, we first create $x\_{t}$, then we train model $f\_{\theta}(x\_{t})$ to predict given $x\_{0}$, using following objective: $CE(f\_\theta(x\_{t}), x\_0)$.
>
> **2: VQ-Diffusion training algorithm**
>
> Given data $x\_{0}$, we first perturb to obtain $x\_{t}$, then we train model $g\_\phi(x\_{t})$ to predict $x\_{0}$ but using different objective:
>
> $KL(q(x\_{t-1}|x\_t, x\_0))||q(x\_{t-1}|x\_t, g\_\phi(x_t))$
>
> We can see that VQ-diffusion is based on the posterior $q(x_{t-1}|x_t, x_0)$ to train the model while the MaskGit, Bert and Paella **does not**. This indicate that the $x_0$ predicted from $g_\phi$ is put under an **additional posterior constraint** while $x_0$ predicted from $f_\theta$ does not. Therefore, if we use the Algorithm 2 (DDPM-inversion) into such models like MaskGit, Bert and Paella, it will **limit the ability of editing controllability** since the $f_\theta$ prediction is not under posterior constraint. Whereas with VQ-diffusion, if using algorithm 1, we **cannot edit** since the noise inversion in algorithm 1 does not take into account the posterior distribution while the pretrained model $g_\phi$ is trained under that constraint.
>
> In summary, we present **two editing algorithm**. While the first one is used for mask generative modelling such as Bert, Maskgit and Paella to **maximize the editing controllability of inverse noise**, the second algorithm is the first editing algorithm extending DDPM-inversion for discrete diffusion. Our paper is not only limited for VQ-diffusion but also notably can be used for mask generative model.
>
> These discussion will be added to the paper to **avoid the confusion about novelty** for reader.
>
> We hope that the above response could address your W1 and please let us know if you have any other concern. We are happy to provide more clarification and positively engage into discussion.

---

> ### Author Response · Authors · 2024-11-28
> **Reminder**
>
> Dear reviewer Vpe1, thank you for your detailed and thoughtful review of our paper. We just want to let you know that we have recently provided the rebuttal answer for weakness 1 (the novelty weakness) and weakness 3 (mutual analysis). We would like to know if our rebuttal is reasonable and well address your concern. If yes, could you please consider to revise your score. Thank you very much for your time and consideration!
>
> If you have any other concerns, we would like to hear and provide clarification.

---

### Author Response · Authors · 2024-11-22

We appreciate the reviewers’ acknowledgment of our contributions. They recognized our method as **novel** (EPpL), serving as the **first inversion method for masked diffusion models** (qRGw,EPpL). Reviewers found our problem formulation and solutions to be **clear and sound** (Vpe1), supported by **clear theoretical foundations**(qRGw) and **well-structured evaluations** (Vpe1). The introduction of a new dataset for semantic text editing was highlighted as a valuable contribution with potential to be **a standardized benchmark** (qRGw). They also appreciated our experiments, noting that they included **sufficient metrics that demonstrate the effectiveness of the methods** (qRGw).

We have addressed the reviewers’ concerns in our rebuttal and are open to further discussions. Thank you for your constructive feedback, which has been instrumental in improving our work\!

---

### Note · Authors · 2024-11-30

I have read and agree with the venue's withdrawal policy on behalf of myself and my co-authors.